# The novel ciliogenesis regulator DYRK2 governs Hedgehog signaling during mouse embryogenesis

**Saishu Yoshida[1], Katsuhiko Aoki[1], Ken Fujiwara[2], Takashi Nakakura[3], Akira Kawamura[1], Kohji Yamada[1], Masaya Ono[4], Satomi Yogosawa[1], Kiyotsugu Yoshida[1]***

[1]Department of Biochemistry, The Jikei University School of Medicine, Tokyo, Japan; [2]Division of Histology and Cell Biology, Department of Anatomy, Jichi Medical University School of Medicine, Tochigi, Japan; [3]Department of Anatomy, Graduate School of Medicine, Teikyo University, Tokyo, Japan; [4]Department of Clinical Proteomics, National Cancer Center Research Institute, Tokyo, Japan

**Abstract** Mammalian Hedgehog (Hh) signaling plays key roles in embryogenesis and uniquely requires primary cilia. Functional analyses of several ciliogenesis-related genes led to the discovery of the developmental diseases known as ciliopathies. Hence, identification of mammalian factors that regulate ciliogenesis can provide insight into the molecular mechanisms of embryogenesis and ciliopathy. Here, we demonstrate that DYRK2 acts as a novel mammalian ciliogenesis-related protein kinase. Loss of *Dyrk2* in mice causes suppression of Hh signaling and results in skeletal abnormalities during in vivo embryogenesis. Deletion of *Dyrk2* induces abnormal ciliary morphology and trafficking of Hh pathway components. Mechanistically, transcriptome analyses demonstrate down-regulation of *Aurka* and other disassembly genes following *Dyrk2* deletion. Taken together, the present study demonstrates for the first time that DYRK2 controls ciliogenesis and is necessary for Hh signaling during mammalian development.

**\*For correspondence:**
kyoshida@jikei.ac.jp

**Competing interests:** The authors declare that no competing interests exist.

## Introduction

Embryogenesis and patterning of cell differentiation are facilitated by spatiotemporal activation of multiple signaling pathways. The Hedgehog (Hh) signaling is an evolutionarily conserved system that plays a central role in embryogenesis via regulating cell proliferation and differentiation (*Ingham and McMahon, 2001*). Upon stimulation by ligands, post-translational modification of GLI2 and GLI3 induces the expression of *Gli1*, which is a key amplifier of Hh signaling. These post-translational and transcriptional activations of three GLIs regulate specific and redundant target genes (*Mo et al., 1997*; *Hui and Angers, 2011*). Hence, mutants of Hh components cause typical defects such as skeletal, neural, and retinal abnormalities (*Mo et al., 1997*).

Unlike other core developmental signaling, vertebrate Hh signaling is uniquely and completely dependent upon primary cilia, which are microtubule-based organelles that are formed during the $G_0$ or $G_1$ phases of the cell cycle (*Huangfu et al., 2003*). Binding of Hh ligands to Patched 1 (PTCH1) on cilia leads to activation and induction of Seven-spanner smoothened (SMO) to the cilia (*Rohatgi et al., 2007*). Activated SMO leads to recruitment of GLI2 and GLI3 to the cilia tip via inhibition of protein kinase A (*Chen et al., 2009*; *Kim et al., 2009*; *Wen et al., 2010*). This dynamic ciliary trafficking of Hh components is primarily regulated by intraflagellar transport (IFT) (*Haycraft et al., 2005*; *Eguether et al., 2014*). Thus, ciliogenesis is indispensable for tissue development, and defects in this process impact the development of multiple organs to cause human and mouse diseases termed 'ciliopathies' (*Reiter and Leroux, 2017*). Accordingly, the typical phenotype

observed in some of ciliopathies such as Joubert syndrome is abnormalities of Hh signaling (*Bangs and Anderson, 2017*).

Mutations in a number of different ciliopathy-associated genes often result in alterations of ciliary length (*Paige Taylor et al., 2016*; *Reiter and Leroux, 2017*). Indeed, genetic screenings of *Chlamydomonas*, which is a model organism for ciliogenesis, have identified the ciliopathy-associated genes controlling cilia length to generate the optimal length at steady-state (*Wemmer and Marshall, 2007*). Although these abnormalities in ciliary length are thought to be controlled by the balance of assembly and disassembly via IFT and a postulated length sensor, the mechanisms for maintaining the cell-type-specific ciliary length have not been fully elucidated (*Ishikawa and Marshall, 2011*). In contrast, the ciliary resorption mechanisms for cell cycle re-entry have been thoroughly investigated by such as an experiment of serum re-addition to starved cells, and these mechanisms include the HEF1-AURKA-HDAC6 pathway (*Pugacheva et al., 2007*), the PLK-KIF2A pathway (*Wang et al., 2013*), and the NEK2-KIF24 pathway (*Kobayashi et al., 2011*; *Kim et al., 2015b*) that have been observed to induce disassembly of cilia. On the other hand, the ability of these ciliary resorption factors for cell cycle re-entry to control cilia length at steady-state and during ciliogenesis remains to be elucidated. Hence, identification of novel mammalian factors regulating ciliogenesis and ciliary length control will provide insight into the molecular mechanisms underlying embryogenesis and ciliopathy as well as ciliary functions.

Dual-specificity tyrosine-regulated kinase (DYRK) is a family that belongs to the CMGC group that includes cyclin-dependent kinases (CDK), mitogen-activated protein kinase (MAPK), glycogen synthase kinase (GSK), and CDK-like kinase (CLKs) (*Becker and Sippl, 2011*). Two isoforms of *Dyrk2* have been identified; long and short forms, the latter lacks a 5' terminal region. In human cancer cells, we have functionally identified DYRK2 as a regulator of p53-induced apoptosis in response to DNA damage (*Taira et al., 2007*) and of G1/S transition (*Taira et al., 2012*). During development in lower eukaryotes, MBK2, which is an ortholog of DYRK2 in *Caenorhabditis elegans*, regulates maternal-protein degradation during the oocyte-to-embryo transition via a ubiquitin-dependent mechanism (*Pelletieri et al., 2003*; *Pang et al., 2004*; *Lu and Mains, 2007*; *Yoshida and Yoshida, 2019*). While these reports lead us to speculate that DYRK2 must also play important roles in mammalian development, no reports are available regarding the mechanistic role of DYRK2 in vivo.

In the present study, we aim to reveal a function for DYRK2 in mammalian development in vivo. Here, we demonstrate that DYRK2 is a novel regulator of ciliogenesis and is required for normal embryogenesis via activation of Hh signaling during development.

## Results

### *Dyrk2* deficiency cause suppression of Hedgehog signaling during mouse embryogenesis

We generated *Dyrk2* knockout mice (*Dyrk2$^{-/-}$*) by eliminating the third exon of the *Dyrk2* genomic locus (*Figure 1—figure supplement 1A–B*). The absence of DYRK2 protein in homozygous *Dyrk2$^{-/-}$* mice was confirmed (*Figure 1—figure supplement 1C*). Although the gross morphology of homozygous *Dyrk2$^{-/-}$* embryos appeared normal during early development, multiple defects became obvious during later stages of gestation, and the mice died at or close to birth (*Figure 1A*). Specifically, defects in skeletal development were remarkable, and these included a shorter dorsum of the nose (*Figure 1A*), cleft palate including hypoplasia of the tongue (*Figure 1B–C*), loss of the basisphenoid, basioccipital, and presphenoid bones (*Figure 1D*), shorter limbs (*Figure 1E*), defects of segmentation of the sternebrae in the sternum (*Figure 1F*), and vertebra (*Figure 1G*) at embryonic day (E) 18.5. These skeletal defects that included reduction of bone mineralization were observed until E16.5 (*Figure 1H*).

As *Dyrk2$^{-/-}$* embryos at E18.5 exhibited a similar phenotype to that observed in response to certain defects in Hh signaling (*Mo et al., 1997*), we assessed *Gli1*-expression, which is an indicator of Hh signaling activation (*Niewiadomski and Rohatgi, 2015*). In situ hybridization demonstrated that *Gli1*-expression was decreased in the craniofacial region in *Dyrk2$^{-/-}$* embryos at E14.5 (*Figure 2A*). Protein levels of GLI1 were also decreased at E13.5 (*Figure 2B*). *Ptch1*-expression, which is another indicator of Hh signaling activation (*Snouffer et al., 2017*), was also decreased in *Dyrk2$^{-/-}$* embryos at E13.5, and this was accompanied by a decrease in *Gli1*; however, *Shh*-expression remained

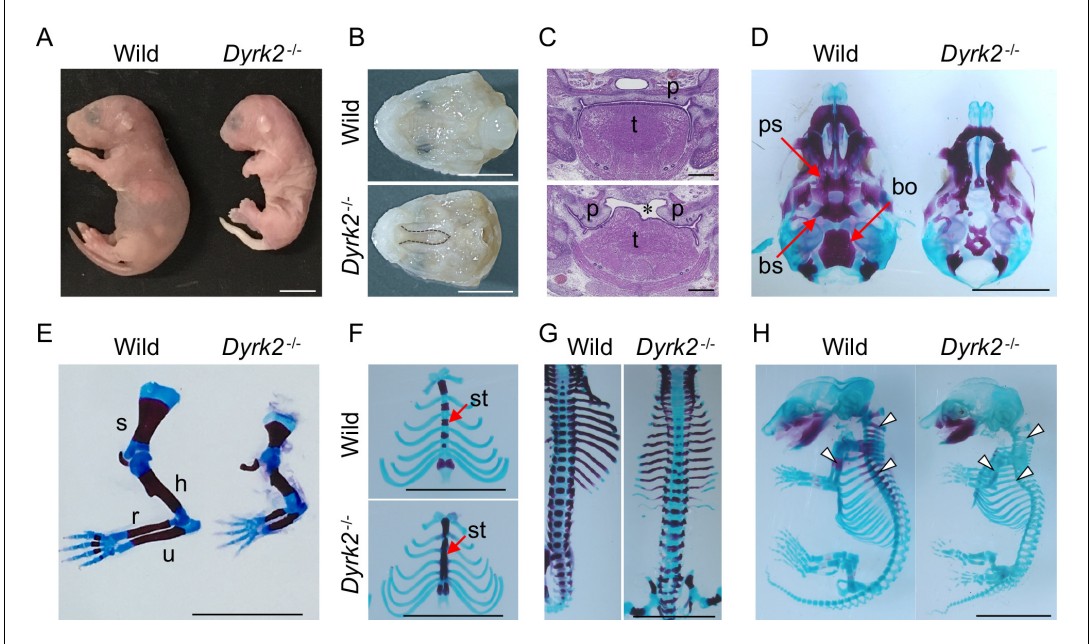

**Figure 1.** Deletion of DYRK2 shows skeletal defects in mouse development. (**A**) Whole embryo gross images of wild-type and homozygous *Dyrk2⁻/⁻* embryos at birth. (**B, C**) Palatal and tongue abnormalities in *Dyrk2⁻/⁻* embryos. Gross images of the palate with mandible removed from wild-type and *Dyrk2⁻/⁻* embryos at E18.5 (**B**), and HE staining from the coronal plane at E13.5 (**C**). Dotted lines in (**B**) and an asterisk in (**C**) indicate cleft of the secondary palate. (**D–H**) Arizarin red and alcian blue staining of the craniofacial skeleton (**D**), forelimbs (**E**), sternum (**F**), and vertebra (**G**) from wild-type and *Dyrk2⁻/⁻* embryos at E18.5, and whole skeleton staining at E16.5 (**H**). Arrowheads in (**H**) indicate regions that decreasing bone mineralization. bo, basioccipital bone; bs, basisphenoid; h, humerus; r, radius; p, palatal shelves; ps, presphenoid; s, scapula; st, sternebrae; t, tongue; u, ulna. Scale bars, 5 mm.

The online version of this article includes the following figure supplement(s) for figure 1:

**Figure supplement 1.** Generation of *Dyrk2⁻/⁻* mice schematic representation of the *Dyrk2⁻/⁻* allele (*Dyrk2*^tm1b).

unchanged (*Figure 2C*). We also observed a repression of *Foxf2*-expression, which is a direct target gene of GLI1 (*Everson et al., 2017*), in the craniofacial region of *Dyrk2⁻/⁻* embryos (*Figure 2D–E*).

Loss of genes required for Hh signaling often causes defects in dorsal-ventral neural tube patterning, which is regulated by the SHH morphogen (*Dessaud et al., 2008*). Although we investigated the localization patterns of FOXA2, NKX2.2, OLIG2, NKX6.1, and PAX6 at E10.5, we did not observe obvious differences in their expression patterns (*Figure 2—figure supplement 1A*). Expression of Hh target genes was decreased in *Dyrk2⁻/⁻* whole embryos at E9.5 (*Figure 2—figure supplement 1B*). In situ hybridization demonstrated that *Ptch1*-expression was decreased in the mandibular arch in *Dyrk2⁻/⁻* embryos at E10.5, but remained unchanged in the neural tube (*Figure 2—figure supplement 1C*). These data showing the maintenance of Hh signal and dorsal-ventral patterning in the neural tube might correspond to a spatiotemporal expression-pattern of *Dyrk2*.

Taken together, *Dyrk2*-deficient embryos exhibit a robust suppression of Hh signaling and possess particular skeletal abnormalities during embryogenesis.

## DYRK2 positively regulates Hh signaling

To investigate the defect in Hh signaling in *Dyrk2⁻/⁻* mice in more detail, we analyzed primary mouse embryonic fibroblasts (MEFs) derived from wild-type and *Dyrk2⁻/⁻* mice. First, we measured Hh signaling activity in response to stimulation with the SMO agonist SAG. In response to stimulation with SAG, *Gli1* and *Ptch1* expression was increased in wild-type mice as previously reported (*Figure 3A*; *Niewiadomski and Rohatgi, 2015*). In contrast, in *Dyrk2⁻/⁻* MEFs, inductions of both *Gli1* and *Ptch1* expression by SAG was drastically repressed (*Figure 3A*). Consistent with gene expression analyses, the induction of GLI1 protein by SAG stimulation was also suppressed in *Dyrk2⁻/⁻* MEFs (*Figure 3B*). Immunocytostaining for GLI1 following stimulation with SAG

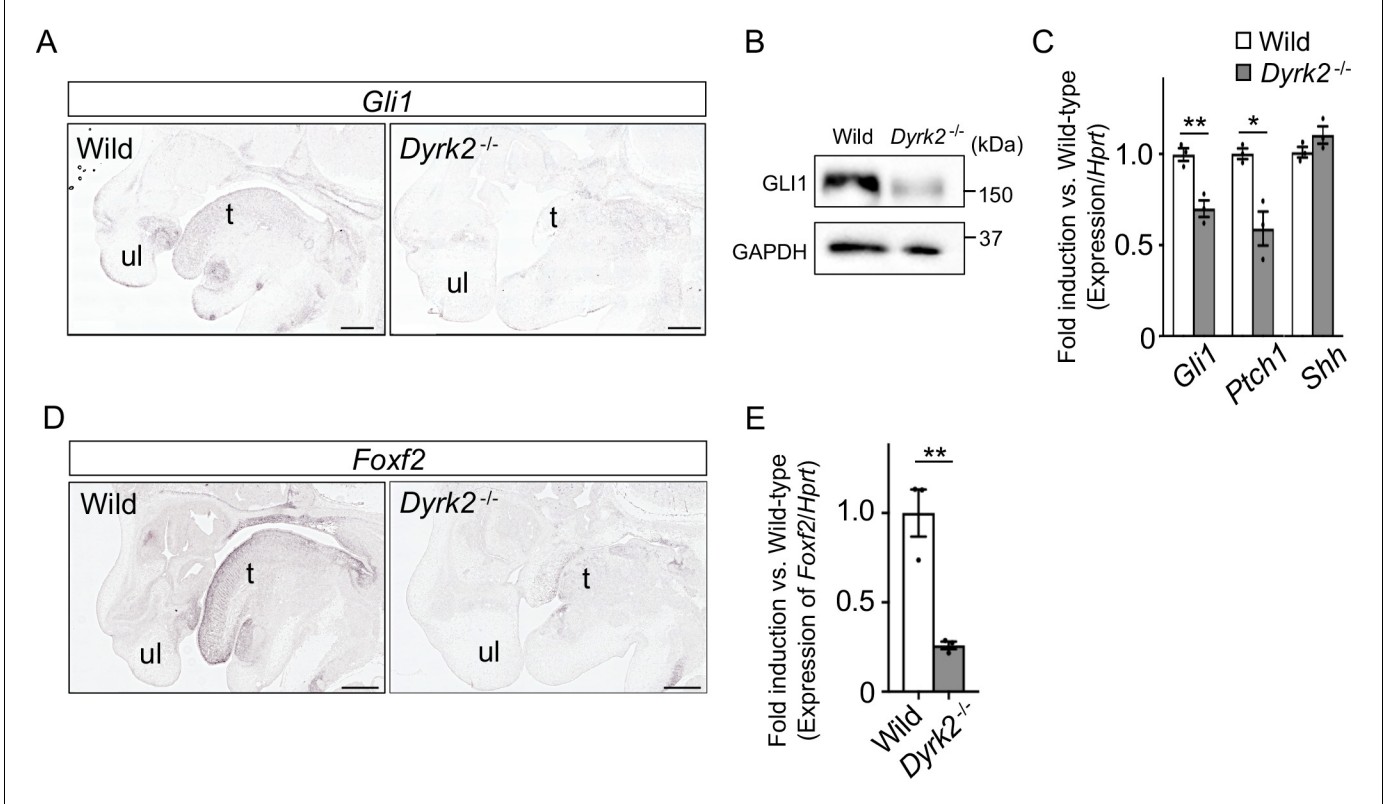

**Figure 2.** Deletion of DYRK2 affects activation of Hh signaling in mouse development. (**A**) In situ hybridization of *Gli1* in the craniofacial region in wild-type and *Dyrk2*<sup>-/-</sup> embryos from the sagittal plane at E14.5. (**B**) Immunoblotting of GLI1 in extracts from the limbs of wild-type and *Dyrk2*<sup>-/-</sup> embryos at E13.5. GAPDH serves as a loading control. (**C**) qPCR of *Gli1*, *Ptch1*, and *Shh* in the limbs from wild-type and *Dyrk2*<sup>-/-</sup> embryos at E13.5. (**D, E**) Repression of *Foxf2*-expression in the craniofacial region of *Dyrk2*<sup>-/-</sup> mice. (**D**) In situ hybridization of *Foxf2* in the craniofacial region in wild-type and *Dyrk2*<sup>-/-</sup> embryos from the sagittal plane at E14.5. (**E**) qPCR of *Foxf2* in the mandibular arch from wild-type and *Dyrk2*<sup>-/-</sup> embryos at E10.5. Hypoxanthine phosphoribosyltransferase (*Hprt*) in (**C and E**) was used as an internal standard, and fold change was calculated by comparing expression levels relative to those of wild-type. Data are presented as the means ± SEM ($n = 3$ biological replicates). The statistical significance between wild-type and *Dyrk2*<sup>-/-</sup> was determined by the Student's *t*-test. (*) $p<0.05$, (**) $p<0.01$. t, tongue; ul, upper lip. Scale bars, 500 μm.

The online version of this article includes the following source data and figure supplement(s) for figure 2:

**Source data 1.** Source data for *Figure 2C and E*.

**Figure supplement 1.** Dorsal-ventral patterning of the neural tube in *Dyrk2*<sup>-/-</sup> mice.

**Figure supplement 1—source data 1.** Source data for *Figure 2—figure supplement 1B*.

demonstrated that the accumulation of GLI1 protein within nuclei was clearly diminished in *Dyrk2*<sup>-/-</sup> MEFs (*Figure 3C*). To validate whether these phenotypes observed in *Dyrk2*<sup>-/-</sup> MEFs are due to abnormal differentiation caused by deletion of *Dyrk2* during early embryogenesis, we performed a transient knockdown of *Dyrk2* in wild-type MEFs using two independent short interfering RNAs (siRNAs). Transient knockdown of *Dyrk2* also demonstrated suppression of both the mRNA and protein levels of *Gli1* and *Ptch1* in response to stimulation with SAG (*Figure 3—figure supplement 1A–B*).

To validate this suppression of Hh signaling by *Dyrk2*-deletion, we performed a transient over-expression experiment using wild-type human *DYRK2* or a *DYRK2-K251R* construct that expresses a kinase dead mutant (*Taira et al., 2012*; *Figure 3—figure supplement 1C–D*) in *Dyrk2*<sup>-/-</sup> MEFs using adenovirus infection (*Yokoyama-Mashima et al., 2019*). Over-expression of the wild-type *DYRK2* construct restored significant induction of *Gli1* and *Ptch1* expression upon exposure to SAG (*Figure 3D*, *Figure 3—figure supplement 1D*). In sharp contrast, over-expression of the *DYRK2-K251R* construct in *Dyrk2*<sup>-/-</sup> MEFs markedly diminished *Gli1* and *Ptch1* expression (*Figure 3D*). Additionally, over-expression of the *DYRK2-K251R* construct slightly increased *Gli1* and *Ptch1* expression in comparison with that of empty vector (*Figure 3D*). This kinase-independent effect might be

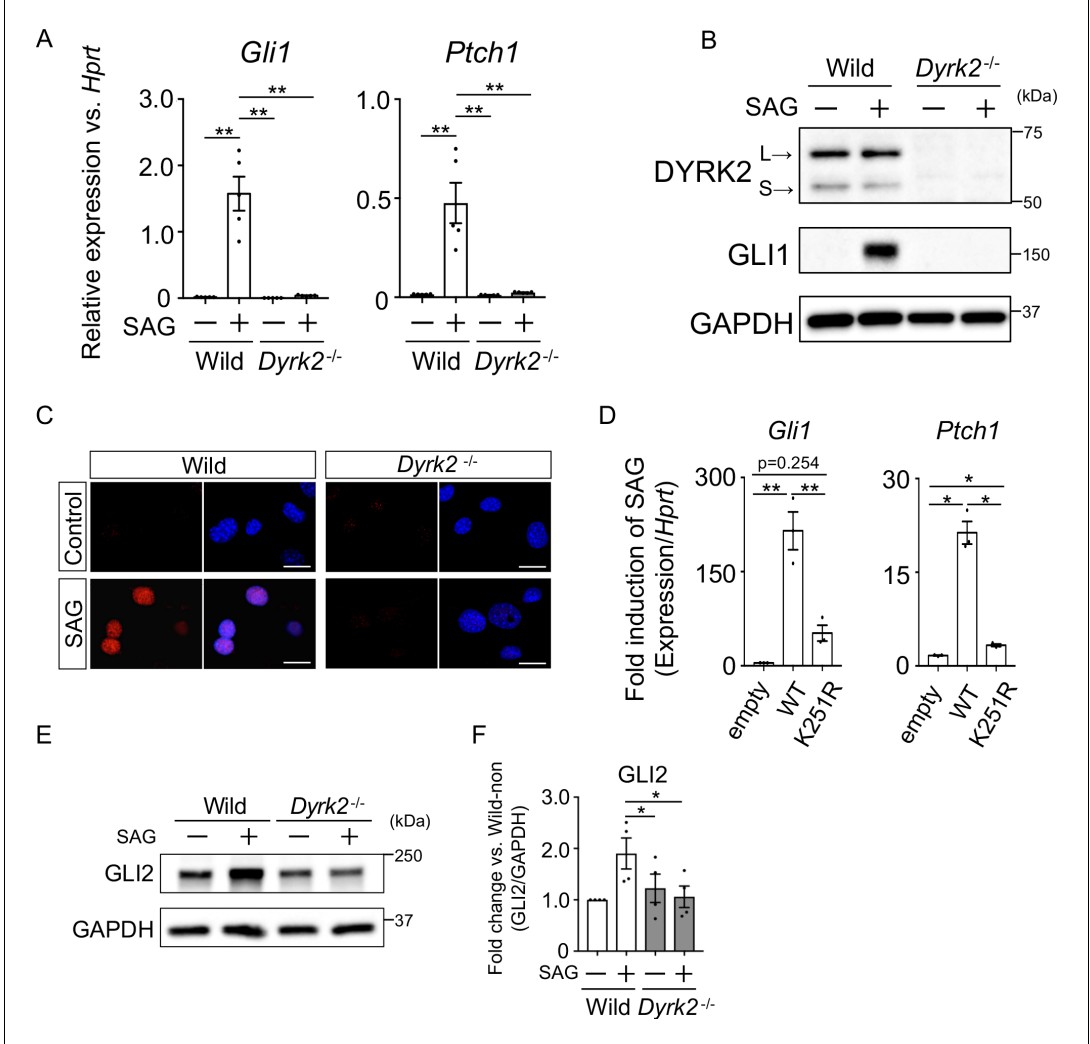

**Figure 3.** Deletion of *Dyrk2* suppresses activation of Hh signaling in vitro. (**A**) Expression of the Hh target genes *Gli1* and *Ptch1* in wild-type and *Dyrk2⁻/⁻* MEFs in the absence or presence of 100 nM SAG was measured by qPCR. Data are shown as relative expression to *Hprt*. (**B**) Protein levels of GLI1 and DYRK2 in wild-type and *Dyrk2⁻/⁻* MEFs in the absence or presence of 100 nM SAG were measured by immuno-blotting. L and S indicate long and short transcriptional isoforms of DYRK2, respectively. (**C**) Wild-type and *Dyrk2⁻/⁻* MEFs in the absence or presence of 100 nM SAG were immune-cytostained for GLI1 (red). Nuclei were stained with DAPI (blue). Scale bars, 5 μm. (**D**) Expression of *Gli1* and *Ptch1* in *Dyrk2⁻/⁻* MEFs overexpressing human *DYRK2* or *DYRK2-K251R* (kinase dead) constructs via adenovirus infection was measured by qPCR. Data indicates fold induction of 100 nM SAG against vehicle after normalization to *Hprt*. (**E, F**) Immunoblotting for GLI2 in wild-type and *Dyrk2⁻/⁻* MEFs in the absence or presence of 100 nM SAG. Protein level as fold changes of GLI2 (**E**) was calculated by comparing protein levels relative to those of wild-type MEFs in the absence of SAG after normalization to the GAPDH loading control in (**F**). Data are presented as the means ± SEM (*n* = 5, 3, and 4 biological replicates per condition in A, D, and F, respectively). The statistical significance was determined by one-way ANOVA followed by Tukey's multiple comparison test. (*) p<0.05, (**) p<0.01. The online version of this article includes the following source data and figure supplement(s) for figure 3:

**Source data 1.** Source data for *Figure 3A and D*.

**Source data 2.** Source data for *Figure 3F*.

**Figure supplement 1.** A transient knockdown of *Dyrk2* suppresses activation of Hh signaling.

**Figure supplement 1—source data 1.** Source data for *Figure 3—figure supplement 1A*.

**Figure supplement 2.** Deletion of *Dyrk2* affects the stabilities of GLI3 Immuno-blotting for GLI3 in wild-type and *Dyrk2⁻/⁻* MEFs in the absence or presence of 100 nM SAG.

**Figure supplement 2—source data 1.** Source data for *Figure 3—figure supplement 2B–D*.

---

associated with a function of DYRK2 as a scaffold protein (*Maddika and Chen, 2009*). Taken together, the induction of Hh signaling is drastically suppressed by deletion of *Dyrk2* in a kinase activity-dependent manner.

The key Hh pathway components GLI2 and GLI3 are known to be posttranslationally modified in a manner that is dependent upon Hh ligands (*Mo et al., 1997*; *Hui and Angers, 2011*). In the absence of Hh ligands, the full-length proteins (active forms; GLI2 and GLI3$^{FL}$) are phosphorylated by multiple kinases, leading to proteasomal degradation or truncation into N-terminal repressor forms (GLI3$^{REP}$), respectively (*Niewiadomski and Rohatgi, 2015*). In this context, we analyzed the endogenous protein levels and states of GLI2 and GLI3. In wild-type MEFs, immunoblotting for GLI2 revealed that full-length form of GLI2 was increased by SAG-stimulation (*Figure 3E–F*). In contrast to wild-type MEFs, the increase of GLI2 protein levels by SAG was significantly suppressed in *Dyrk2*$^{-/-}$ MEFs (*Figure 3E–F*). We also analyzed two forms of GLI3 (GLI3$^{FL}$ and GLI3$^{REP}$). SAG-stimulation suppressed the formation of GLI3$^{REP}$ and decreased the ratio of GLI3$^{REP}$/GLI3$^{FL}$ in wild-type MEFs (*Figure 3—figure supplement 2*; *Niewiadomski and Rohatgi, 2015*). In *Dyrk2*$^{-/-}$ MEFs, however, we found that the ratio of GLI3$^{REP}$/GLI3$^{FL}$ was marginally suppressed and that no significant differences between the absence or presence of SAG existed (*Figure 3—figure supplement 2D*). Collectively, these data indicate that the deletion of *Dyrk2* affects the stabilities of GLI2, and marginally GLI3, under SAG-stimulation.

## DYRK2 regulates ciliogenesis

As primary cilia are essential organelles required for signal transduction of vertebrate Hh signaling (*Huangfu et al., 2003*), we investigated whether DYRK2 regulates ciliogenesis in MEFs. Immunostaining of acetylated tubulin (a cilia axoneme marker) and γ-tubulin (a basal body marker) demonstrated that the length of primary cilia in *Dyrk2*$^{-/-}$ MEFs was significantly longer than that in wild-type MEFs (*Figure 4A*). The average cilia length in wild-type MEFs was $1.65 \pm 0.03$, while it was $3.59 \pm 0.08$ in *Dyrk2*$^{-/-}$ MEFs (*Figure 4B–C*). In addition to the increased length, the morphology of primary cilia in *Dyrk2*$^{-/-}$ MEFs was often bulged at the tips, tapered, and twisted (*Figure 4A*). This elongation and morphological abnormality of primary cilia was similarly observed in response to the transient knockdown of *Dyrk2* by siRNA in wild-type MEFs (*Figure 4—figure supplement 1*). To investigate whether the regulation of ciliogenesis by DYRK2 is conserved in other species and cell types, we analyzed immortalized human retinal pigment epithelia cells (hTERT-RPE1 cells) that are commonly used to study cilia-assembly and -disassembly. A transient knockdown by si*DYRK2* also induced a significant elongation and morphological abnormality in cilia of these cells (*Figure 4—figure supplement 2*). In contrast to the length and morphology of primary cilia, no difference was observed on the proportion of ciliated cells in wild-type and *Dyrk2*$^{-/-}$ MEFs (*Figure 4—figure supplement 3A–B*). Similarly, in cell-cycling (KI67-positive) wild-type and *Dyrk2*$^{-/-}$ MEFs, there was comparable in the proportion of ciliated cells (ciliated cells in KI67-positive cells is 1 per 199 and 1 per 139 cells in wild-type and *Dyrk2*$^{-/-}$ MEFs, respectively) (*Figure 4—figure supplement 3C*).

To confirm the morphological abnormalities of primary cilia within the tissue, we performed scanning electron microscopy (SEM) on embryos at E10.5. SEM images clearly showed that the cilia of *Dyrk2*$^{-/-}$ embryos were significantly elongated, bulged at the tips, and twisted, while those of wild-type embryos were shortened and straight (*Figure 4D*). These abnormalities were also observed in several types of cells, including mesenchymal cells (*Figure 4E*), chondrocytes, neuroepithelium, and tongue cells (data not shown) in the embryonic craniofacial region at E13.5 as assessed by immunohistochemistry.

These data indicating that deletion of *Dyrk2* causes morphological abnormalities in primary cilia prompted us to determine the subcellular localization of DYRK2. We transfected DYRK2-HaloTag constructs into hTERT-RPE1 cells and induced ciliogenesis by serum-starvation. Immunocytostaining for both anti-HaloTag and anti-DYRK2 revealed that DYRK2 localized at γ-tubulin-positive basal bodies and at the proximal end of the axoneme, namely a transition zone (TZ) (*Figure 5A–B*). No signal for anti-HaloTag (*Figure 5C*) or anti-DYRK2 (data not shown) was observed in hTERT-RPE1 cells transfected with empty vector (pFN22K-Halo Tag-CMVd1-Flexi-vector). Moreover, immuno-positive signals for DYRK2-HaloTag were co-localized with a TZ marker, NPHP1 (*Figure 5D*).

These data indicated that DYRK2 might regulate ciliogenesis at basal bodies and TZ in vivo and in vitro.

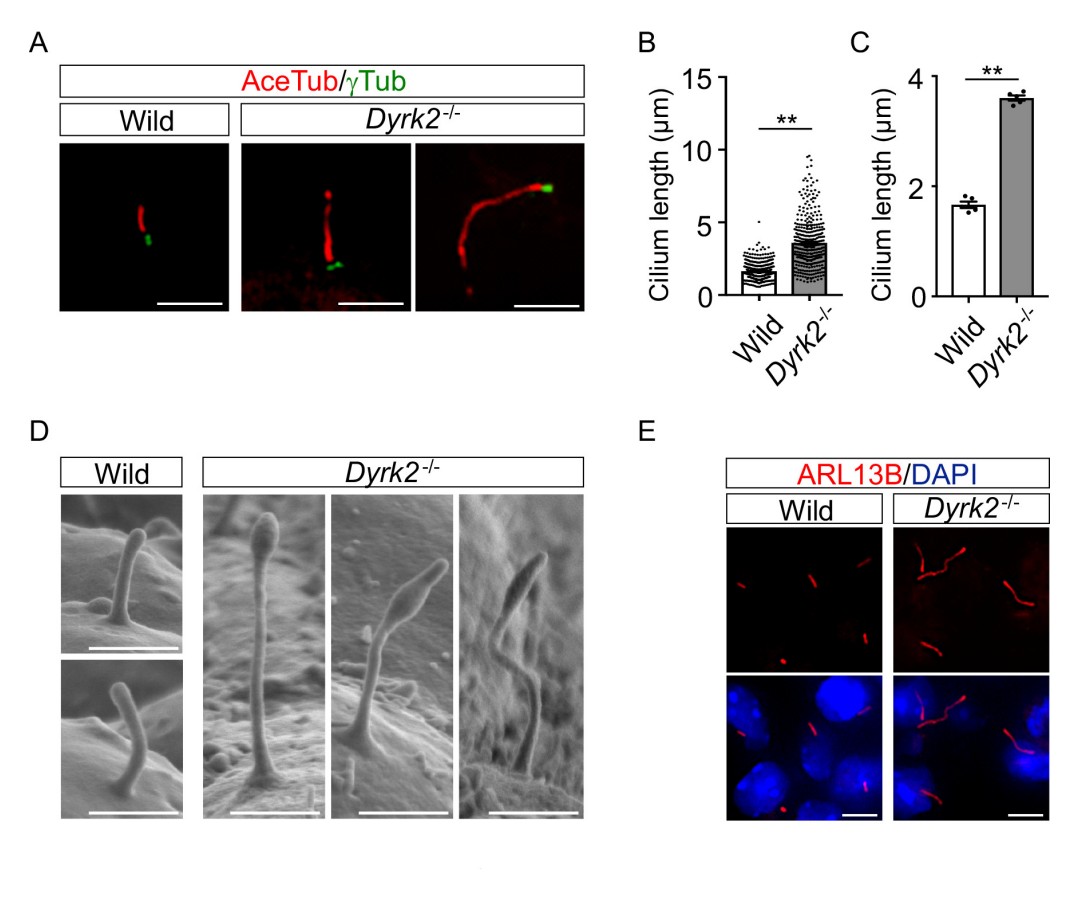

**Figure 4.** DYRK2 constrains the length of primary cilia. (**A–C**) Elongation of primary cilia in *Dyrk2⁻/⁻* MEFs. Primary cilia of wild-type and *Dyrk2⁻/⁻* MEFs were immunostained with acetylated-tubulin and gamma-tubulin antibodies. (**B, C**) Measurements of cilia length in wild-type and *Dyrk2⁻/⁻* MEFs using acetylated-tubulin as a cilia axoneme marker. Cilia lengths are presented as pooled from five MEFs derived from independent embryos of each genotype (**B**) and the average of each MEF (**C**). Data are presented as the means ± SEM (*n* = 5 biological replicates per condition). The statistical significance between wild-type and *Dyrk2⁻/⁻* was determined by the Student's *t*-test. (**\*\***) p<0.01. (**D**) Scanning electron microscopy showing wild-type and *Dyrk2⁻/⁻* embryos in the frontonasal prominence at E10.5. (**E**) Immunohistochemistry of primary cilia in wild-type and *Dyrk2⁻/⁻* embryos. ARL13B was immuno-stained in wild-type and *Dyrk2⁻/⁻* mesenchymal cells at the craniofacial region at E13.5. Nuclei were stained with DAPI. Scale bars, 5 μm (**A** and **E**) and 1 μm (**D**).

The online version of this article includes the following source data and figure supplement(s) for figure 4:

**Source data 1.** Source data for *Figure 4B–C*.
**Figure supplement 1.** Elongation of primary cilia in wild-type MEFs treated with si*Dyrk2*.
**Figure supplement 1—source data 1.** Source data for *Figure 4—figure supplement 1B–C*.
**Figure supplement 2.** Elongation of primary cilia in hTERT-RPE1 cells treated with si*DYRK2*.
**Figure supplement 2—source data 1.** Source data for *Figure 4—figure supplement 2A and C–D*.
**Figure supplement 3.** Quantification of the proportion of ciliated cells in wild-type and *Dyrk2⁻/⁻* MEFs.
**Figure supplement 3—source data 1.** Source data for *Figure 4—figure supplement 3B*.

## Deletion of DYRK2 induces abnormal ciliary trafficking of Hedgehog pathway components

In mammals, key regulators of Hh signaling have been demonstrated to be recruited and activated at the cilia upon Hh stimulation (*Chen et al., 2009*; *Kim et al., 2009*; *Tukachinsky et al., 2010*; *Wen et al., 2010*), and disorders in the ciliary trafficking of Hh components cause dysfunction of Hh signaling (*He et al., 2014*). To investigate whether inactivation of Hh signaling in *Dyrk2⁻/⁻* embryos and MEFs is due to abnormal ciliary trafficking of Hh components, we analyzed the ciliary localization of key regulators such as SMO, GLI2, GLI3, and SuFu. In wild-type MEFs, immuno-positive SMO signals were mostly undetectable or faint in the absence of Hh stimulation, and recruitment to cilia was

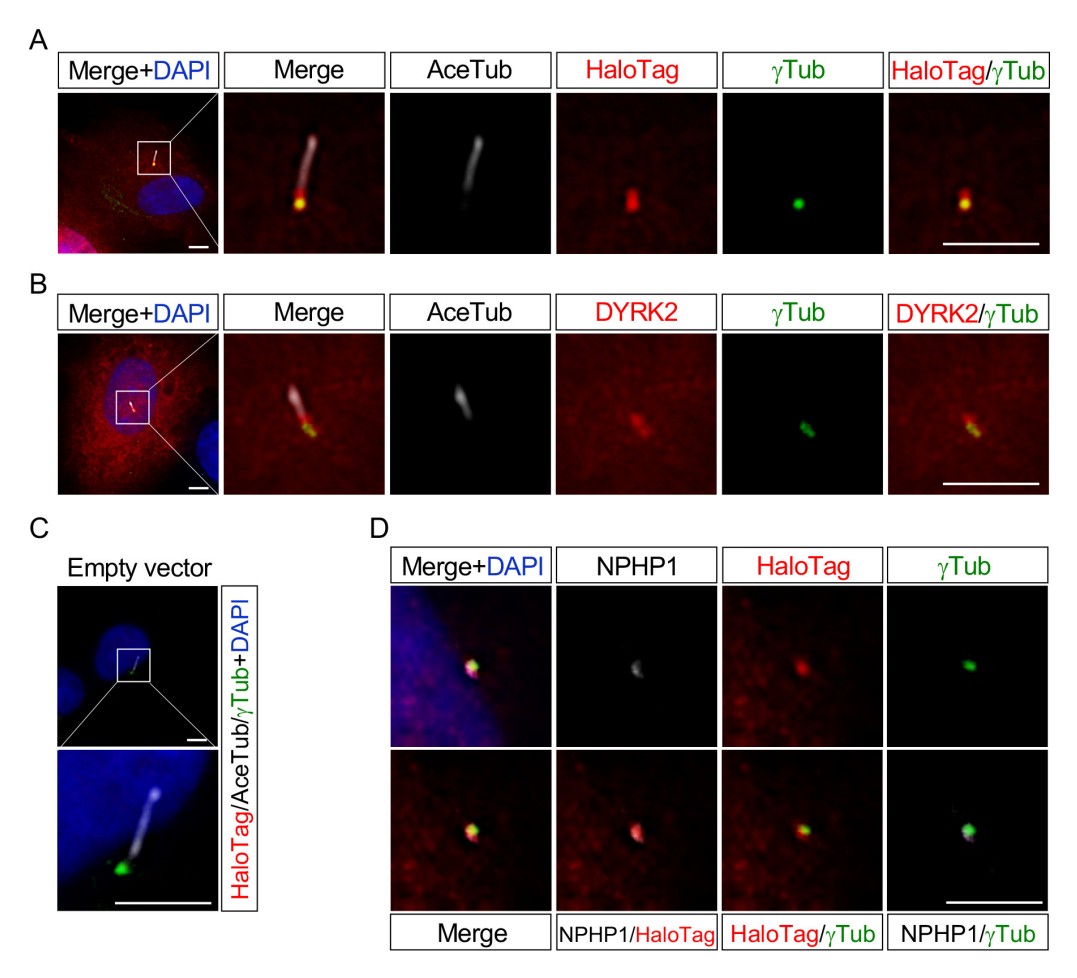

**Figure 5.** DYRK2 localizes at basal bodies and transition zone (TZ) in primary cilia. Cultured hTERT-RPE1 cells were transfected with a mouse DYRK2-HaloTag overexpression construct and immunostained using anti-HaloTag (**A**) or anti-DYRK2 (**B**) with acetylated-tubulin (white) and gamma-tubulin antibodies. (**C**) Cultured hTERT-RPE1 cells transfected with an empty vector (pFN22K-Halo Tag-CMVd1-Flexi-vector) and immunostained using anti-HaloTag with acetylated-tubulin (white) and gamma-tubulin antibodies. (**D**) Co-localization of DYRK2 and a TZ marker, NPHP1. Cultured hTERT-RPE1 cells overexpressed with a mouse DYRK2-HaloTag were immunostained using anti-HaloTag, NPHP1 (white), and gamma-tubulin antibodies. Nuclei were stained with DAPI. Scale bars, 5 μm.

dependent upon Hh stimulation (*Figure 6A–B*), as shown in a previous report (*Tukachinsky et al., 2010*). Similarly, no significant difference in the frequency of SMO recruitment in response to SAG stimulation was observed in *Dyrk2*[-/-] MEFs (*Figure 6A–B*).

We subsequently analyzed the ciliary localization of endogenous GLI2 and GLI3. As shown in a previous report (*Tukachinsky et al., 2010*), in wild-type MEFs, immuno-positive signals for both GLI2 and GLI3 were mostly undetectable or faint in the absence of Hh stimulation, and SAG stimulation increased the presence of GLI2 and GLI3 at cilia tips (*Figure 6C–F*). In contrast, in *Dyrk2*[-/-] MEFs, immuno-positive signals for both GLI2 (approximately 75.9% of cilia) and GLI3 (approximately 95.8%) were observed at cilia tip even in the absence of SAG treatment (*Figure 6C–F*). Moreover, the intensity of immune-positive signals was markedly increased by SAG-treatment in *Dyrk2*[-/-] MEFs (*Figure 6C,E*). These accumulations at cilia tips were frequently observed in *Dyrk2*[-/-] mesenchymal cells in the craniofacial region at E10.5 tissues but were absent in wild-type cells (*Figure 6—figure supplement 1*). Additionally, localization of SuFu, which forms a complex with both GLI2 and GLI3 (*Tukachinsky et al., 2010*), was disordered in cilia tips in a similar pattern to that of GLI2 and GLI3 (*Figure 6—figure supplement 2A*). Collectively, ciliary localization of GLI2, GLI3, and SuFu in *Dyrk2*[-/-] MEFs and embryos was clearly disordered and was accumulated at cilia tips. Importantly,

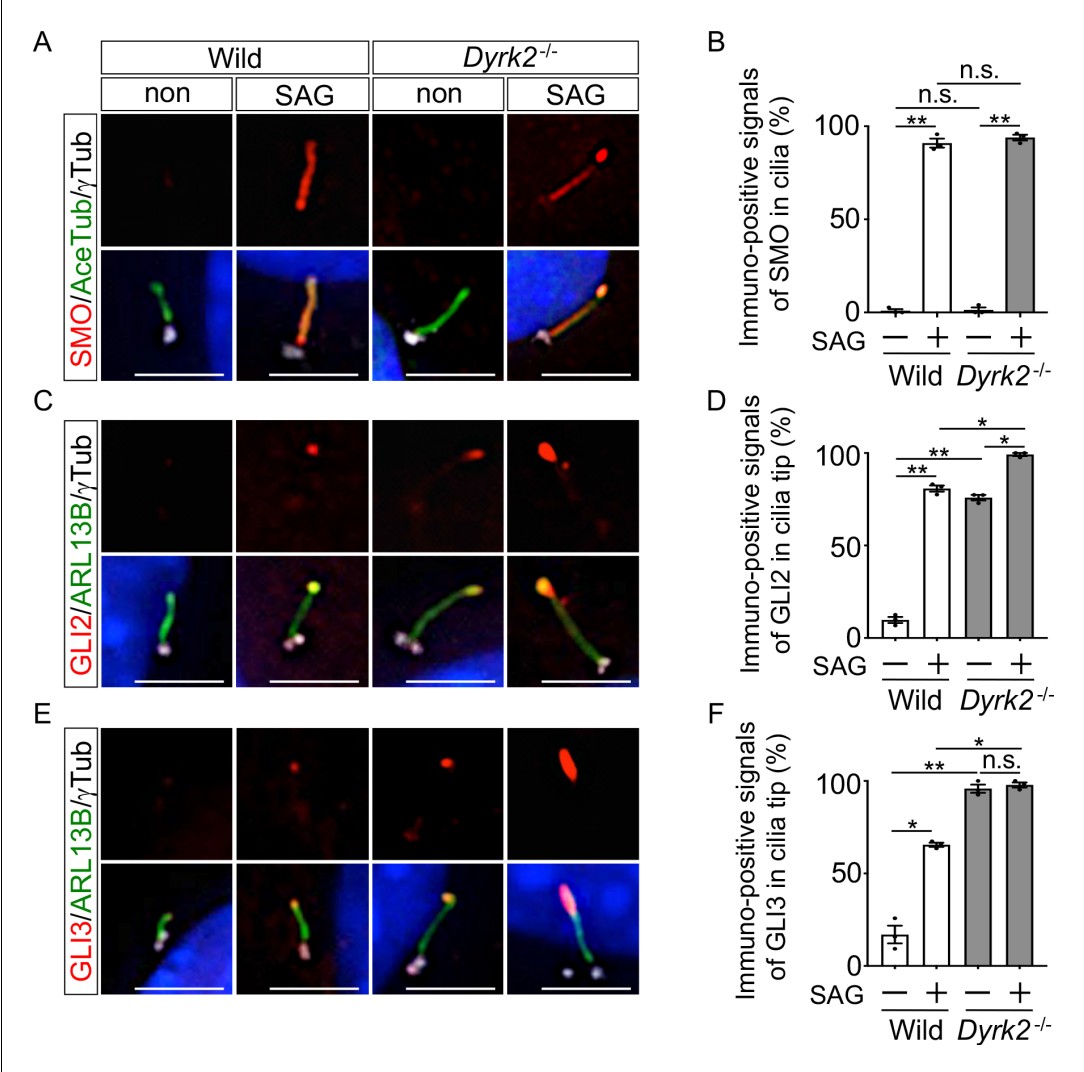

**Figure 6.** Depletion of *Dyrk2* induces abnormal ciliary trafficking of endogenous Hh components. Ciliary localization of endogenous SMO, GLI2, and GLI3 in wild-type and *Dyrk2⁻/⁻* MEFs in the absence or presence of 100 nM SAG. Primary cilia were immuno-stained for SMO (A), GLI2 (C), or GLI3 (E) with ARL13B and gamma-tubulin (white) antibodies. Nuclei were stained with DAPI (blue). The percentage of cells with SMO (B) at the cilia or foci of GLI2 (D) or GLI3 (F) at the cilia tips was determined. Data are presented as the means ± SEM (*n* = 3 biological replicates for each condition;>110 cells were scored for each experiment). The statistical significance was determined by one-way ANOVA followed by Tukey's multiple comparison test. (*) p<0.05, (**) p<0.01. Scale bars, 5 μm.

The online version of this article includes the following source data and figure supplement(s) for figure 6:

**Source data 1.** Source data for *Figure 6B,D and F*.

**Figure supplement 1.** Depletion of *Dyrk2* induces abnormal ciliary trafficking of endogenous GLI2 and GLI3 in vivo.

**Figure supplement 2.** Immunocytochemistry of endogenous SuFu and IFTs.

**Figure supplement 3.** Effects of rapamycin treatment on cilia.

**Figure supplement 3—source data 1.** Source data for *Figure 6—figure supplement 3C–D*.

**Figure supplement 4.** Protein levels of CP110 and KATANIN p60 in *Dyrk2⁻/⁻* MEFs.

the recruitment of GLI2, GLI3, and SuFu in response to SAG stimulation was also observed in *Dyrk2⁻/⁻* MEFs.

Inhibition of retrograde transport from the tip to the cell body induces accumulation of Hh components and results in abnormal localization of both IFT-A (implicated in retrograde IFT) and IFT-B (implicated in anterograde) (*Ocbina and Anderson, 2008*; *Ocbina et al., 2011*; *Liem et al., 2012*). Based on this, we analyzed the ciliary localization of core IFT-A (IFT140) and IFT-B (IFT81 and IFT88)

(*Nakayama and Katoh, 2018*). In *Dyrk2^-/-* MEFs, no obvious differences in ciliary localization of IFT140, IFT81, and IFT88 were observed (*Figure 6—figure supplement 2B–D*).

Activation of mammalian target of rapamycin complex 1 (mTORC1) also induces abnormal trafficking and elongates cilia length (*Broekhuis et al., 2014*). mTORC1 activation leads to phosphorylation of ribosomal S6 kinase (S6K) and eukaryotic translational initiation factor 4E binding protein (4EBP). In *Dyrk2^-/-* MEFs, phosphorylation of both S6K and 4EBP was slightly increased (*Figure 6—figure supplement 3A*); however, treatment with rapamycin, an inhibitor of mTORC1, resulted in no obvious differences in cilia length in *Dyrk2^-/-* MEFs (*Figure 6—figure supplement 3B–D*).

Moreover, a centrosome protein CP110 (*Hossain et al., 2017*) and a microtubule severing enzyme, KATANIN p60 (*Maddika and Chen, 2009*), have been identified as substrates of DYRK2 for proteolysis. In *Dyrk2^-/-* MEFs, however, no obvious difference in protein levels of both CP110 and KATANIN p60 was observed (*Figure 6—figure supplement 4*).

### Deletion of *Dyrk2* dysregulates the expression of *Aurka* and other cilia-disassembly genes

To understand the molecular mechanisms underlying cilia dysfunction in *Dyrk2^-/-* mice, we focused on factors that are involved in ciliary length control by incorporating whole-genome RNA sequencing using wild-type and *Dyrk2^-/-* MEFs (*Figure 7*, *Figure 7—figure supplement 1*). The data were analyzed by multiple testing and according to p-value, false discovery rate (FDR), and ratio (*Dyrk2^-/-*/wild-type). As a result, the number of identified genes was 53 or 42 that were significantly downregulated (p<0.005, ratio <1.5 fold) or upregulated (p<0.005, ratio >1.5 fold) in *Dyrk2^-/-* MEFs, respectively, regardless of the presence or absence of SAG (*Table 1*). Notably, GO and STRING analysis revealed that the 53 downregulated genes in *Dyrk2^-/-* MEFs were enriched in cell division (GO: 0051301, FDR = 5.17E-40), microtubule cytoskeleton organization (GO:0000226, FDR = 5.23E-15), spindle organization (GO:0007051, FDR = 2.72E-11), mitotic cell cycle checkpoint (GO:0007093, FDR = 3.78E-07), and microtubule-based movement (GO:0007018, FDR = 3.9E-4) (*Figure 7A*, *Figure 7—figure supplement 1B*). These downregulated genes in *Dyrk2^-/-* MEFs included those related to ciliary resorption mechanisms for proliferation, including the HEF1-AURKA-HDAC6 pathway (*Aurka*, *Plk1*, *Ube2c*, and *Tpx2*) (*Pugacheva et al., 2007*), the PLK-KIF2A pathway (*Plk1* and *Kif2c*, a family of *Kif2a*) (*Wang et al., 2013*), and the APC^CDC20-Nek1 pathway (*Cdc20*) that controls ciliary length (*Wang et al., 2014*; *Keeling et al., 2016*). We confirmed the downregulation of *Aurka*, *Plk1*, *Ube2c*, *Tpx2*, *Kif2c*, and *Cdc20* in *Dyrk2^-/-* MEFs by qPCR (*Figure 7B*). To identify a molecule involved in cilia-elongation in *Dyrk2^-/-* cells, we performed transient knockdown of selected genes using siRNA in wild-type MEFs, and we analyzed cilia length. Notably, we found that knockdown of *Aurka* by two independent siRNAs significantly increases cilia length (*Figure 8*). Moreover, we performed a rescue experiment by over-expression of AURKA-EGFP in *Dyrk2^-/-* MEFs (*Figure 9*). Immunostaining and measurement of the cilia length in EGFP- (transfected with pEGFP-C1) or AURKA-EGFP-positive (transfected with *Aurka*/pEGFP-C1) *Dyrk2^-/-* MEFs demonstrated that elongated cilia in *Dyrk2^-/-* MEFs were significantly shortened in AURKA-EGFP-positive cells in comparison with EGFP-positive ones (*Figure 9D–G*).

These findings collectively support the potential mechanism that DYRK2 governs ciliogenesis by, at least in part, maintaining the expression of *Aurka* and other disassembly genes.

## Discussion

### DYRK2 is a positive regulator of Hh signaling

Although evidence indicates that DYRK2 plays important roles in the development of lower eukaryotes (*Pellettieri et al., 2003*; *Pang et al., 2004*; *Lu and Mains, 2007*), little is known regarding the functions of DYRK2 in mammalian development. In the present study, we demonstrate for the first time that DYRK2 is required for normal Hh signaling and embryogenesis in vivo. Varjosalo et al. have established a human full-length protein kinase cDNA and corresponding kinase activity-deficient mutant library, and they reported that DYRK2 functions as a negative regulator of Hh signaling via direct phosphorylation and induction of the proteasome-dependent degradation of GLIs using in vitro over-expression approaches (*Varjosalo et al., 2008*). In sharp contrast, our present study demonstrated using knockout approaches that endogenous protein levels of GLI2, and marginally the

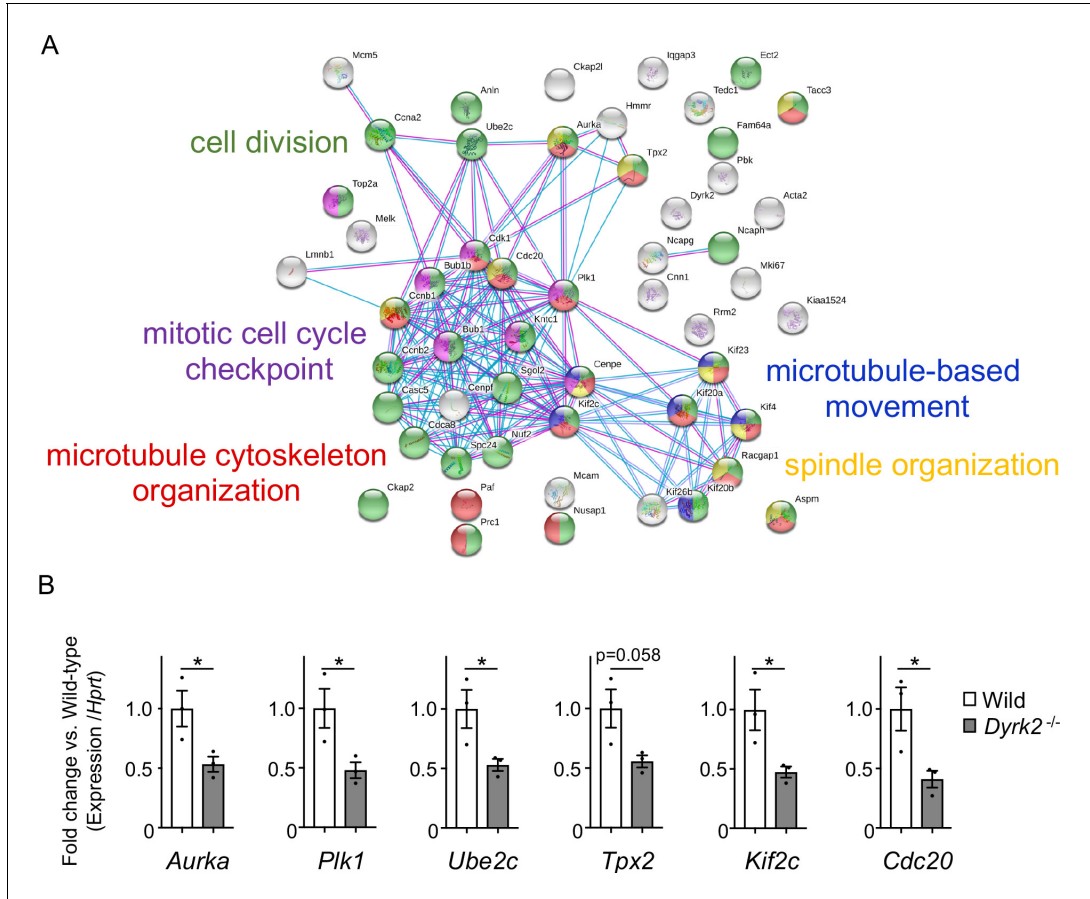

**Figure 7.** Changes in mRNA expression of genes in *Dyrk2*$^{-/-}$ MEFs. (**A**) STRING GO analyses of the 53 differentially downregulated genes in *Dyrk2*$^{-/-}$ MEFs reveals protein-protein interaction networks. Robust networks for cell division (green, GO: 0051301), microtubule cytoskeleton organization (red, GO:0000226), spindle organization (yellow, GO:0007051), mitotic cell cycle checkpoint function (purple, GO:0007093), and microtubule-based movement (blue, GO:0007018) were extracted. (**B**) Confirmation of downregulation of genes related to ciliary resorption mechanisms in *Dyrk2*$^{-/-}$ MEFs by qPCR. *Hprt* was used as an internal standard, and fold change was calculated by comparing expression levels relative to those of wild-type. Data are presented as the means ± SEM (*n* = 3 biological replicates per condition). The statistical significance between wild-type and *Dyrk2*$^{-/-}$ MEFs was determined using the Student's *t*-test. (*) p<0.05.

The online version of this article includes the following source data and figure supplement(s) for figure 7:

**Source data 1.** Source data for *Figure 7B*.

**Figure supplement 1.** Transcriptome analysis in *Dyrk2*$^{-/-}$ MEFs.

ratio of GLI3$^{REP}$/GLI3$^{FL}$, are decreased by deletion of *Dyrk2* in vitro after SAG-treatment. Consistently, *Dyrk2*$^{-/-}$ embryos exhibited some typical phenotypes of inactivation of Hh signaling in vivo as indicated by abnormal responses but not the elimination of ligands. Taken together with the observed loss of *Gli1* induction in *Dyrk2*$^{-/-}$ MEFs in vitro, we concluded that DYRK2 functions as a positive regulator of Hh signaling. GLI2 and GLI3, the key mediators of Hh signaling, are known to have specific and redundant functions (**Mo et al., 1997**). In skeletal patterning during development, *Dyrk2*$^{-/-}$ embryos exhibit phenotypes that are more similar to those of double *Gli2* and *Gli3* mutants than to those of each single mutant. Conversely, the deletion of other indispensable upstream Hh components such as *Smo* (**Norman et al., 2009**), *Ptch1* (**Svärd et al., 2006**), *Gpr161* (**Hwang et al., 2018**), and *SuFu* (**Svärd et al., 2006**) results in more severe defects that occur at earlier embryonic stages. The present data do not rule out the possibility that DYRK2 directly regulates Hh components. Despite this, given the evidence that *Dyrk2*$^{-/-}$ embryos and cells possess morphological abnormalities in primary cilia, it is clear that DYRK2 plays a pivotal role in regulating Hh signaling via the control of ciliogenesis.

**Table 1.** A list of downregulated or upregulated genes in Dyrk2$^{-/-}$ MEFs

**Down-regulated genes in Dyrk2$^{-/-}$**

| ID | GeneSymbol | Description | Ratio of Dyrk2$^{-/-}$ per wild-type in the presence of SAG | Ratio of Dyrk2$^{-/-}$ per wild-type in the absence of SAG |
|---|---|---|---|---|
| ENSMUSG00000028630 | Dyrk2 | Dual-specificity tyrosine-(Y)-phosphorylation regulated kinase 2 | 0.02 | 0.03 |
| ENSMUSG00000035683 | Melk | Maternal embryonic leucine zipper kinase | 0.23 | 0.22 |
| ENSMUSG00000074476 | Spc24 | NDC80 kinetochore complex component%2C homolog (*S. cerevisiae*) | 0.25 | 0.21 |
| ENSMUSG00000020808 | Pimreg | PICALM interacting mitotic regulator | 0.28 | 0.28 |
| ENSMUSG00000033952 | Aspm | Abnormal spindle microtubule assembly | 0.31 | 0.25 |
| ENSMUSG00000026683 | Nuf2 | NDC80 kinetochore complex component | 0.31 | 0.30 |
| ENSMUSG00000037466 | Tedc1 | Tubulin epsilon and delta complex 1 | 0.31 | 0.26 |
| ENSMUSG00000030867 | Plk1 | Polo-like kinase 1 | 0.31 | 0.17 |
| ENSMUSG00000022033 | Pbk | PDZ binding kinase | 0.33 | 0.29 |
| ENSMUSG00000027326 | Knl1 | Kinetochore scaffold 1 | 0.33 | 0.20 |
| ENSMUSG00000041431 | Ccnb1 | Cyclin B1 | 0.33 | 0.26 |
| ENSMUSG00000036777 | Anln | Anillin actin binding protein | 0.33 | 0.26 |
| ENSMUSG00000001403 | Ube2c | Ubiquitin-conjugating enzyme E2C | 0.33 | 0.25 |
| ENSMUSG00000027496 | Aurka | Aurora kinase A | 0.34 | 0.26 |
| ENSMUSG00000001349 | Cnn1 | Calponin 1 | 0.34 | 0.31 |
| ENSMUSG00000032218 | Ccnb2 | Cyclin B2 | 0.34 | 0.28 |
| ENSMUSG00000026039 | Sgo2a | Shugoshin 2A | 0.34 | 0.25 |
| ENSMUSG00000015880 | Ncapg | Non-SMC condensin I complex subunit G | 0.34 | 0.34 |
| ENSMUSG00000027379 | Bub1 | BUB1 mitotic checkpoint serine/threonine kinase | 0.36 | 0.23 |
| ENSMUSG00000040084 | Bub1b | BUB1B mitotic checkpoint serine/threonine kinase | 0.36 | 0.29 |
| ENSMUSG00000045328 | Cenpe | Centromere protein E | 0.36 | 0.22 |
| ENSMUSG00000032254 | Kif23 | Kinesin family member 23 | 0.37 | 0.25 |
| ENSMUSG00000028873 | Cdca8 | Cell division cycle associated 8 | 0.37 | 0.30 |
| ENSMUSG00000032135 | Mcam | Melanoma cell adhesion molecule | 0.37 | 0.29 |
| ENSMUSG00000027469 | Tpx2 | TPX2microtubule-associated | 0.37 | 0.33 |
| ENSMUSG00000028678 | Kif2c | Kinesin family member 2C | 0.37 | 0.24 |
| ENSMUSG00000027715 | Ccna2 | Cyclin A2 | 0.38 | 0.23 |
| ENSMUSG00000048327 | Ckap2l | Cytoskeleton associated protein 2-like | 0.39 | 0.23 |
| ENSMUSG00000040204 | Pclaf | PCNA clamp associated factor | 0.40 | 0.19 |
| ENSMUSG00000029414 | Kntc1 | Kinetochore associated 1 | 0.42 | 0.24 |
| ENSMUSG00000034311 | Kif4 | Kinesin family member 4 | 0.42 | 0.24 |

*Table 1 continued on next page*

| ID | GeneSymbol | Description | | |
|---|---|---|---|---|
| ENSMUSG00000031004 | Mki67 | Antigen identified by monoclonal antibody Ki 67 | 0.42 | 0.21 |
| ENSMUSG00000020914 | Top2a | Topoisomerase (DNA) II alpha | 0.42 | 0.21 |
| ENSMUSG00000033031 | Cip2a | Cell proliferation regulating inhibitor of protein phosphatase 2A | 0.42 | 0.32 |
| ENSMUSG00000035783 | Acta2 | Actin alpha two smooth muscle aorta | 0.43 | 0.48 |
| ENSMUSG00000024795 | Kif20b | Kinesin family member 20B | 0.43 | 0.30 |
| ENSMUSG00000038943 | Prc1 | Protein regulator of cytokinesis 1 | 0.43 | 0.26 |
| ENSMUSG00000026494 | Kif26b | Kinesin family member 26B | 0.43 | 0.25 |
| ENSMUSG00000023015 | Racgap1 | Rac GTPase-activating protein 1 | 0.43 | 0.26 |
| ENSMUSG00000026605 | Cenpf | Centromere protein F | 0.44 | 0.25 |
| ENSMUSG00000027306 | Nusap1 | Nucleolar and spindle associated protein 1 | 0.45 | 0.28 |
| ENSMUSG00000028068 | Iqgap3 | IQ motif containing GTPase activating protein 3 | 0.46 | 0.21 |
| ENSMUSG00000003779 | Kif20a | Kinesin family member 20A | 0.47 | 0.25 |
| ENSMUSG00000005410 | Mcm5 | Minichromosome maintenance complex component 5 | 0.47 | 0.26 |
| ENSMUSG00000034906 | Ncaph | Non-SMC condensin I complex subunit H | 0.47 | 0.27 |
| ENSMUSG00000006398 | Cdc20 | Cell division cycle 20 | 0.48 | 0.29 |
| ENSMUSG00000037313 | Tacc3 | Transforming acidic coiled-coil containing protein 3 | 0.48 | 0.36 |
| ENSMUSG00000027699 | Ect2 | ect2 oncogene | 0.48 | 0.26 |
| ENSMUSG00000020330 | Hmmr | Hyaluronan-mediated motility receptor (RHAMM) | 0.50 | 0.28 |
| ENSMUSG00000020649 | Rrm2 | Ribonucleotide reductase M2 | 0.50 | 0.26 |
| ENSMUSG00000019942 | Cdk1 | Cyclin-dependent kinase 1 | 0.50 | 0.34 |
| ENSMUSG00000024590 | Lmnb1 | Lamin B1 | 0.51 | 0.33 |
| ENSMUSG00000037725 | Ckap2 | Cytoskeleton associated protein 2 | 0.55 | 0.42 |

**Upregulated genes in Dyrk2$^{-/-}$**

| ID | GeneSymbol | Description | Ratio of Dyrk2$^{-/-}$ per wild-type in the presence of SAG | Ratio of Dyrk2$^{-/-}$ per wild-type in the absence of SAG |
|---|---|---|---|---|
| ENSMUSG00000056673 | Kdm5d | Lysine (K)-specific demethylase 5D | Inf | Inf |
| ENSMUSG00000068457 | Uty | Ubiquitously transcribed tetratricopeptide repeat gene Y chromosome | Inf | Inf |
| ENSMUSG00000069049 | Ddx3y | DEAD (Asp-Glu-Ala-Asp) box polypeptide 3 Y-linked | Inf | 8278 |
| ENSMUSG00000069045 | Eif2s3y | Eukaryotic translation initiation factor 2 subunit three structural gene Y-linked | Inf | Inf |
| ENSMUSG00000112616 | Gm47434 | Predicted gene 47434 | 719 | Inf |
| ENSMUSG00000025582 | Nptx1 | Neuronal pentraxin 1 | 4.74 | 11.91 |
| ENSMUSG00000024164 | C3 | Complement component 3 | 4.47 | 11.59 |
| ENSMUSG00000039457 | Ppl | Periplakin | 4.30 | 11.11 |
| ENSMUSG00000025784 | Clec3b | C-type lectin domain family three member b | 3.99 | 8.60 |

*Table 1 continued on next page*

| ENSMUSG00000002944 | Cd36 | CD36 molecule | 3.20 | 3.45 |
|---|---|---|---|---|
| ENSMUSG00000035385 | Ccl2 | Chemokine (C-C motif) ligand 2 | 2.86 | 2.84 |
| ENSMUSG00000095478 | Gm9824 | Predicted pseudogene 9824 | 2.60 | 4.14 |
| ENSMUSG00000038642 | Ctss | Cathepsin S | 2.58 | 3.19 |
| ENSMUSG00000043719 | Col6a6 | Collagen type VI alpha 6 | 2.44 | 4.64 |
| ENSMUSG00000033327 | Tnxb | Tenascin XB | 2.37 | 3.61 |
| ENSMUSG00000069516 | Lyz2 | Lysozyme 2 | 2.30 | 3.08 |
| ENSMUSG00000016494 | Cd34 | CD34 antigen | 2.29 | 2.26 |
| ENSMUSG00000042129 | Rassf4 | Ras association (RalGDS/AF-6) domain family member 4 | 2.29 | 3.43 |
| ENSMUSG00000004730 | Adgre1 | Adhesion G-protein-coupled receptor E1 | 2.27 | 2.49 |
| ENSMUSG00000030144 | Clec4d | C-type lectin domain family member d | 2.26 | 3.74 |
| ENSMUSG00000029816 | Gpnmb | Glycoprotein (transmembrane) nmb | 2.22 | 2.66 |
| ENSMUSG00000042286 | Stab1 | Stabilin 1 | 2.18 | 2.70 |
| ENSMUSG00000020120 | Plek | Pleckstrin | 2.18 | 2.99 |
| ENSMUSG00000040254 | Sema3d | Sema domain immunoglobulin domain (Ig) short basic domain secreted (semaphorin) 3D | 2.17 | 2.89 |
| ENSMUSG00000005268 | Prlr | Prolactin receptor | 2.17 | 4.44 |
| ENSMUSG00000024621 | Csf1r | Colony-stimulating factor one receptor | 2.10 | 2.74 |
| ENSMUSG00000074896 | Ifit3 | Interferon-induced protein with tetratricopeptide repeats 3 | 2.04 | 3.96 |
| ENSMUSG00000002985 | Apoe | Apolipoprotein E | 2.03 | 2.51 |
| ENSMUSG00000057137 | Tmem140 | Transmembrane protein 140 | 2.02 | 3.18 |
| ENSMUSG00000002289 | Angptl4 | Angiopoietin-like 4 | 2.02 | 5.94 |
| ENSMUSG00000050335 | Lgals3 | Lectin galactose binding soluble 3 | 1.99 | 2.66 |
| ENSMUSG00000090877 | Hspa1b | Heat-shock protein 1B | 1.98 | 2.13 |
| ENSMUSG00000054404 | Slfn5 | Schlafen 5 | 1.96 | 3.77 |
| ENSMUSG00000031209 | Heph | Hephaestin | 1.92 | 2.48 |
| ENSMUSG00000027996 | Sfrp2 | Secreted frizzled-related protein 2 | 1.91 | 5.68 |
| ENSMUSG00000050953 | Gja1 | Gap junction protein alpha 1 | 1.90 | 2.45 |
| ENSMUSG00000005413 | Hmox1 | Heme oxygenase 1 | 1.90 | 1.97 |
| ENSMUSG00000046805 | Mpeg1 | Macrophage expressed gene 1 | 1.85 | 2.57 |
| ENSMUSG00000022037 | Clu | Clusterin | 1.83 | 3.06 |
| ENSMUSG00000026389 | Steap3 | STEAP family member 3 | 1.81 | 2.24 |
| ENSMUSG00000041577 | Prelp | Proline arginine-rich end leucine-rich repeat | 1.81 | 2.01 |
| ENSMUSG00000027339 | Rassf2 | Ras association (RalGDS/AF-6) domain family member 2 | 1.80 | 2.72 |

## *Dyrk2* is a novel ciliogenesis-related gene in mice

Intriguingly, we found that DYRK2 negatively regulates ciliogenesis. In *Chlamydomonas*, certain mutations that cause flagella to assemble to excessive length (i.e. negatively regulator of ciliogenesis) have been identified, and these include mutations in LF1 through LF5 (*Wilson et al., 2008*;

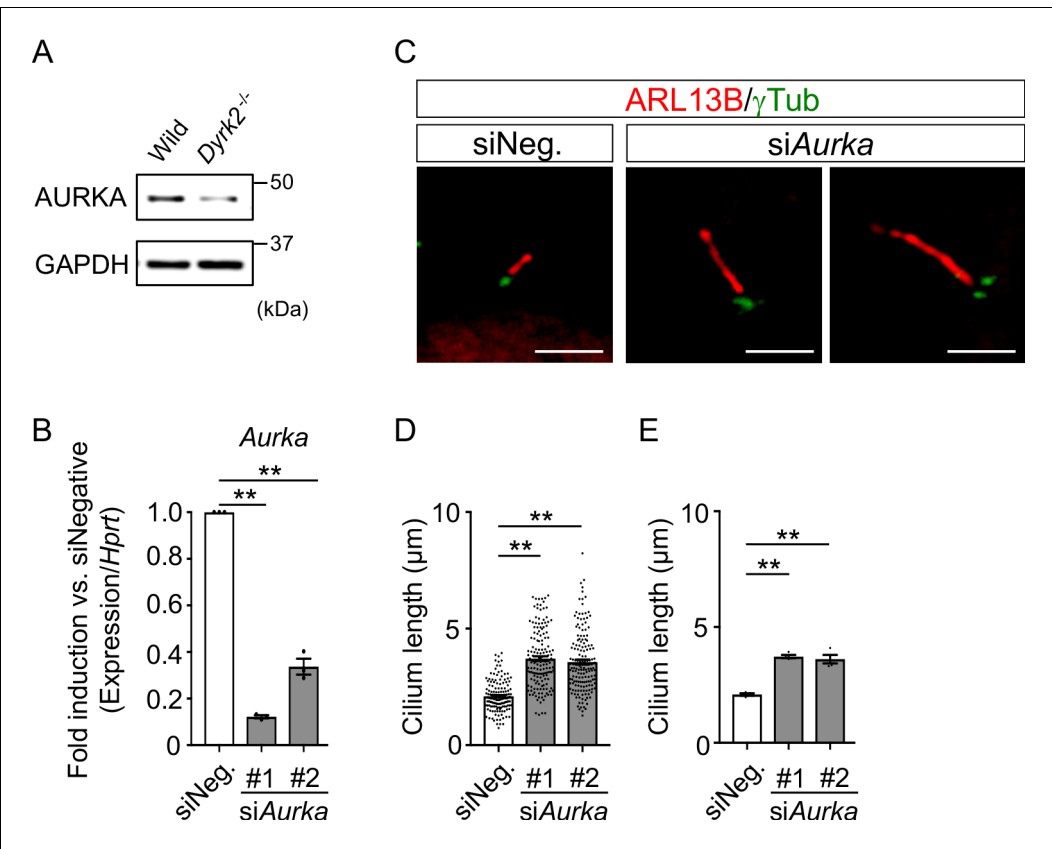

**Figure 8.** Elongation of primary cilia in wild-type MEFs treated with si*Aurka*. (A) Immunoblotting of AURKA in wild-type and *Dyrk2⁻/⁻* MEFs. GAPDH serves as a loading control. (B) Knockdown efficiency of *Aurka*-expression in wild-type MEFs treated with two independent si*Aurka* for 48 hr was measured by qPCR. *Hprt* was used as an internal standard, and fold change was calculated by comparing expression levels relative to those of siNegative (siNeg.). Data are presented as the means ± SEM (*n* = 3 biological replicates per condition). (C) Primary cilia in wild-type cells treated with siNegative (siNeg.) or two independent si*Aurka* were immuno-stained with ARL13B and gamma-tubulin antibodies. Scale bars, 5 μm. (D, E) Measurements of cilia length in wild-type MEFs treated with siNeg. or two independent si*Aurka* using ARL13B and acetylated-tubulin as a cilia axoneme marker. Cilia lengths are presented as pooled from four MEFs derived from independent wild-type embryos (D) and represent an average of each MEF (E). Data are presented as the means ± SEM (*n* = 4 biological replicates per condition). The statistical significance was determined by one-way ANOVA followed by Tukey's multiple comparison test. (**) p<0.01. The online version of this article includes the following source data for figure 8:

**Source data 1.** Source data for *Figure 8B and D–E*.

*Tam et al., 2013*). Among these, LF2, LF4, and LF5 encode protein kinases and are homologs of vertebrate CCRK, MAK/ICK/MOK, and CDKL5, respectively. In addition to these genes, GSK3β also negatively regulates cilia and flagella length (*Yuan et al., 2012*). Interestingly, DYRK2 belongs to the same kinase group as CCRK, MAK/ICK/MOK, CDKL5, and GSK3β (the CMGC group) (*Becker and Sippl, 2011*). To the best of our knowledge, however, the present study demonstrates for the first time that DYRK2 is a ciliogenesis-related gene and a negative regulator of ciliogenesis. Among these kinases, *Dyrk2⁻/⁻* embryos exhibit similar phenotypes to *Ick*-deletion mice in vivo such as skeletal defects and cilia morphology, although they do not possess a hydrocephalus defect (*Moon et al., 2014*). In response to *Ick*- or *Mok*-knockdown, cilia length depends on the activation of mTORC1 signaling (*Bolton et al., 2007*). Our previous study reported the activation of mTORC1 signaling in *DYRK2*-knockdown human breast cancer cells (*Mimoto et al., 2017*). As expected, mTORC1 signaling was slightly activated in *Dyrk2⁻/⁻* MEFs; however, rapamycin, an inhibitor of mTORC1, did not significantly affect cilia length in *Dyrk2⁻/⁻* MEFs. Thus, DYRK2 may control ciliogenesis through different mechanisms than those of other CMGC-kinases.

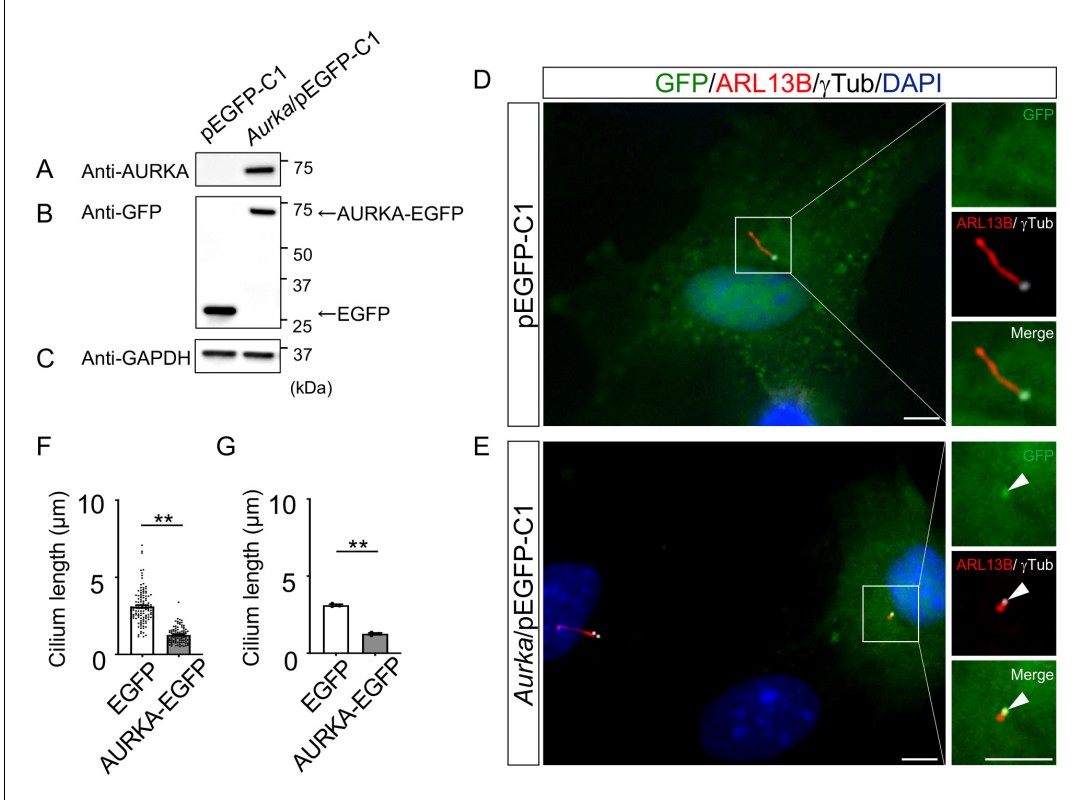

**Figure 9.** Reduction of the length of primary cilia in *Dyrk2*-/- MEFs by over-expression of AURKA. (**A–C**) Immunoblotting by anti-AURKA (**A**), anti-GFP (**B**), and anti-GAPDH (**C**) in cells transfected with pEGFP-C1 or mouse *Aurka*/pEGFP-C1. GAPDH serves as a loading control. (**D, E**) Primary cilia in *Dyrk2*-/- MEFs over-expressed with EGFP (**D**) or AURKA-EGFP (**E**) were immunostained with GFP, ARL13B, and gamma-tubulin (white) antibodies. Arrowheads in (**E**) indicate signals for AURKA-EGFP in gamma-tubulin-positive basal body. Scale bars, 5 μm. (**F, G**) Measurements of cilia length in EGFP- or AURKA-EGFP-over-expressed *Dyrk2*-/- MEFs using ARL13B as a cilia axoneme marker. Cilia lengths in EGFP- or AURKA-EGFP-positive cells are presented as pooled from three MEFs derived from independent *Dyrk2*-/- embryos (**F**) and represent an average of each MEF (**G**). Data are presented as the means ± SEM (*n* = 3 biological replicates per condition). The statistical significance between EGFP- and AURKA-EGFP-positive cells was determined by the Student's *t*-test. (**\*\***) p<0.01.
The online version of this article includes the following source data for figure 9:

**Source data 1.** Source data for *Figure 9F and G*.

## DYRK2 is required for ciliogenesis and the dynamic trafficking of Hedgehog components in cilia

Abnormal ciliary trafficking of Hh components causes dysfunction in Hh signaling in several types of mutant mice (*He et al., 2014*; *Moon et al., 2014*). *Dyrk2*-/- MEFs and embryos possessed disordered accumulation of Hh components (GLI2, GLI3, and SuFu) at cilia tips and exhibited elongation of cilia. On the other hand, SMO recruitment dependent on ligand stimulation was normally observed in *Dyrk2*-/- MEFs (*Figure 10*). Moreover, DYRK2 localizes at TZ, which acts as a selective barrier to control ciliary import and export of proteins (*Gerhardt et al., 2016*). Given the findings that ciliary localization and ciliary disorders were observed in *Dyrk2*-/- cells, DYRK2 could be involved in regulation of ciliary protein entry and exit. The functions of DYRK2 at TZ, however, remains to be elucidated.

As a first step to elucidate the functions of DYRK2 in cilia, we focused on factors involved in ciliary length control using a transcriptome approach, and we identified the downregulation of genes related to ciliary resorption mechanisms for proliferation in *Dyrk2*-/- MEFs, such as *Aurka*, *Plk1*, *Ube2c*, *Tpx2*, *Kif2c,* and *Cdc20*. Among these, AURKA has been well-characterized as a disassembly factor, as transient knockdown by siRNA or treatment with an inhibitor of AURKA completely blocked ciliary disassembly during proliferation (*Pugacheva et al., 2007*; *Inoko et al., 2012*). In contrast to ciliary disassembly under proliferation, the function of AURKA for controlling ciliary length at steady state is unclear. In the present study, transient knockdown of *Aurka* under serum-starvation

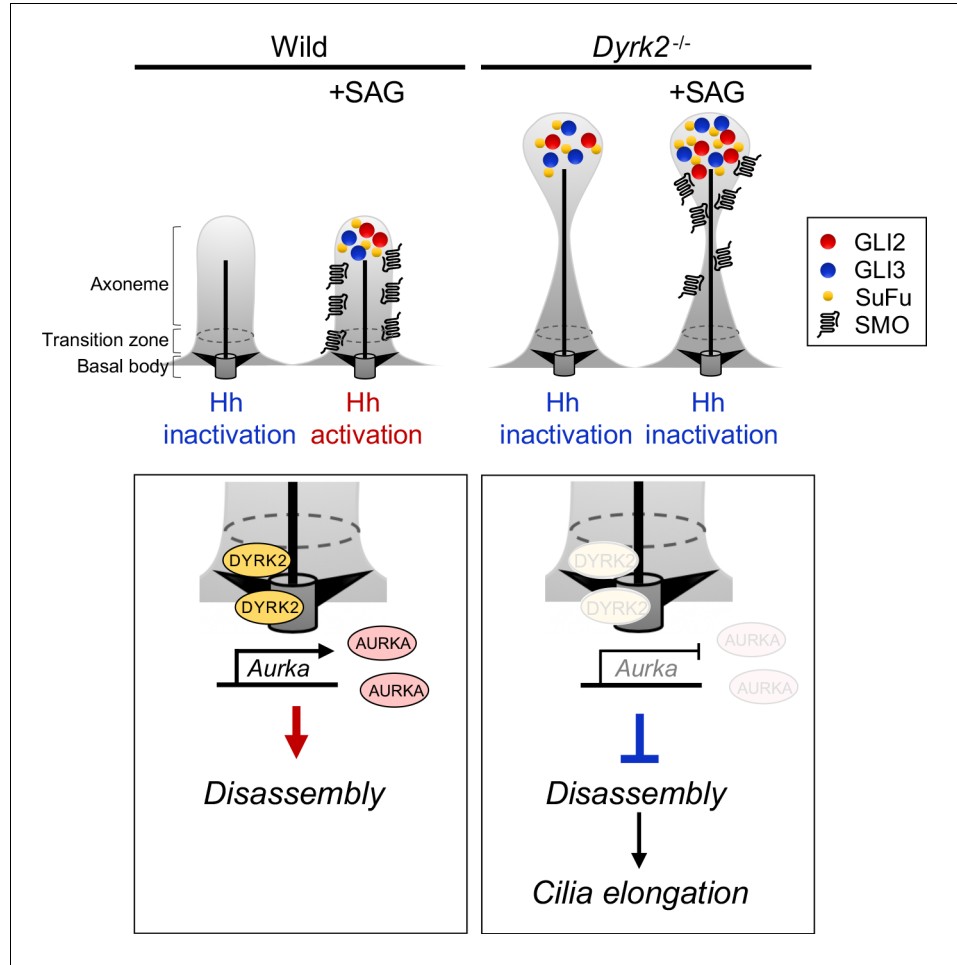

**Figure 10.** Schematic representation of DYRK2 in ciliogenesis and Hh signaling. (Left panel) A schematic model of normal ciliogenesis and response to stimulation with Hh ligand. (Right panel) A schematic model ciliogenesis and response to stimulation with Hh ligand in *Dyrk2*-deletion. The morphology of primary cilia in *Dyrk2*$^{-/-}$ MEFs was elongated and often bulged at the tips. In *Dyrk2*$^{-/-}$ cells, downregulation of *Aurka* and other ciliary disassembly genes caused suppression of disassembly and elongation of primary cilia. Furthermore, abnormal ciliary trafficking caused accumulation of GLI2, GLI3, and SuFu in *Dyrk2*$^{-/-}$ cells. Consequently, the induction of Hh signaling is drastically suppressed by deletion of *Dyrk2*.

conditions induced elongation of cilia in a manner similar to that observed in *Dyrk2*$^{-/-}$ MEFs. These data imply that the down-regulation of *Aurka* is, at least in part, associated with the phenotypes observed after deletion of *Dyrk2*. The expression of *Aurka* is known to be regulated by pathways such as YAP/TAZ (*Kim et al., 2015a*) and AKT signaling (*Plotnikova et al., 2015*). The molecular mechanisms underlying DYRK2-mediated *Aurka* regulation and ciliary trafficking remain unclear. Further research is required to elucidate these mechanisms.

## A possible relationship between DYRK2 and human ciliopathy

A number of syndromes caused by disorders involving ciliary proteins are categorized as skeletal ciliopathies, and these include short-rib thoracic dysplasia (SRTD), Jeune asphyxiating thoracic dystrophy (JATD), orofaciodigital syndrome (OFD), Ellis-van Creveld syndrome (EVC), and cranioectodermal dysplasia (CED) (*Reiter and Leroux, 2017*). In mice, deletion of *Dyrk2* induces morphological abnormalities in primary cilia and skeletal defects in vivo. Additionally, DYRK2 is a ciliary protein that is primarily localized at the basal body and the TZ, which contains a growing number of ciliopathy proteins (*Reiter et al., 2012*). While the present study does not include any evidence to

support the relationship between DYRK2 and human disease, our results do suggest a possibility that *DYRK2* is involved in human ciliopathy, particularly in regard to skeletal disorders. Further investigations involving exome sequencing or genome-wide association studies using human patients will prove useful to clarify this issue.

## Conclusion

In summary, we identified DYRK2 as a novel mammalian ciliogenesis-related gene in vivo and in vitro. Deletion of *Dyrk2* induces abnormal ciliary morphology and trafficking of Hh pathway components and suppresses Hh signaling during mouse embryogenesis. The abnormal ciliogenesis in *Dyrk2*$^{-/-}$ cells is partially caused by downregulation of *Aurka* and other disassembly genes. These findings will allow for a more complete understanding of the molecular mechanisms underlying embryogenesis, ciliogenesis, and human ciliopathy.

# Materials and methods

## Key resources table

| Reagent type (species) or resource | Designation | Source or reference | Identifiers | Additional information |
|---|---|---|---|---|
| Genetic reagent (*M. musculus*) | *Dyrk2*$^{-/-}$ mouse | This paper | N/A | Maintained in K. Yoshida lab. |
| Cell line (*M. musculus*) | Wild-type and *Dyrk2*$^{-/-}$ MEFs | This paper | N/A | Maintained in K. Yoshida lab. |
| Cell line (*H. sapiens*) | hTERT-RPE1 | ATCC | Cat# CRL-4000 RRID:CVCL_4388 | |
| Transfected construct (*M. musculus*) | mouse *Aurka*/ pEGFP-C1 | This paper | N/A | See Materials and methods subsection 'Plasmid constructs' |
| Transfected construct (*M. musculus*) | mouse *Dyrk2*/ FN22K-Halo Tag-CMVd1-Flexi-vector | This paper | N/A | See Materials and methods subsection 'Plasmid constructs' |
| Transfected construct (*M. musculus*) | *Dyrk2* targeting vector | Knockout Mouse Project Repository | PG00105_X_1_G09, PG00105_X_1_E04 | See Materials and methods subsection 'Plasmid constructs' |
| Recombinant DNA regent | Plasmid pEGFP-C1 (empty vector) | TaKaRa Bio | Cat# 6084–1 | |
| Recombinant DNA regent | Plasmid pFN22K-Halo Tag-CMVd1-Flexi-vector (empty vector) | Promega | Cat# G2851 | |
| Transfected construct (*M. musculus*) | *Dyrk2* targeting vector | Knockout Mouse Project Repository | PG00105_X_1_G09, PG00105_X_1_E04 | |

*Continued on next page*

*Continued*

| Reagent type (species) or resource | Designation | Source or reference | Identifiers | Additional information |
|---|---|---|---|---|
| Biological sample (Adenovirus) | Adenovirus-*Cre* | *Yokoyama-Mashima et al., 2019* doi: 10.1016/j.canlet.2019.02.046. | N/A | |
| Biological sample (Adenovirus) | Adenovirus-human *DYRK2* | *Yokoyama-Mashima et al., 2019* doi: 10.1016/j.canlet.2019.02.046. | N/A | |
| Biological sample (Adenovirus) | Adenovirus-human *DYRK2-K251R* | *Yokoyama-Mashima et al., 2019* doi: 10.1016/j.canlet.2019.02.046. | N/A | |
| Biological sample (Adenovirus) | Adenovirus-GFP | *Yokoyama-Mashima et al., 2019* doi: 10.1016/j.canlet.2019.02.046. | N/A | |
| Antibody | Anti-Acetylated-tubulin (Mouse monoclonal) | Sigma-Aldrich | Cat# T7451, RRID:AB_609894 | ICC (1:2000) |
| Antibody | Anti-ARL13B (Mouse monoclonal) | Abcam | Cat# ab136648, N/A | ICC (1:300) |
| Antibody | Anti-ARL13B (Rabbit polyclonal) | Proteintech | Cat# 17711–1-AP, RRID:AB_2060867 | ICC (1:400) / IHC (1:400) |
| Antibody | Anti-AURKA (Mouse monoclonal) | BD Transduction | Cat# 610938, RRID:AB_398251 | WB (1:1000) |
| Antibody | Anti-DYRK2 (Rabbit polyclonal) | Sigma-Aldrich | Cat# HPA027230, RRID:AB_1847925 | WB (1:1000) / ICC (1:400) |
| Antibody | Anti-FOXA2 (Mouse monoclonal) | Developmental Studies Hybridoma Bank | Cat# 4C7, RRID:AB_528207 | IHC (1:8) |
| Antibody | Anti-CP110 (Rabbit polyclonal) | Proteintech | Cat# 12780–1-AP, RRID:AB_10638480 | WB (1:1000) |
| Antibody | Anti-GAPDH (Mouse monoclonal) | Santa Cruz Biotechnology | Cat# sc-32233, RRID:AB_627679 | WB (1:3000) |
| Antibody | Anti-GFP (Chicken polyclonal IgY) | Aves Labs | Cat# GFP-1020, RRID:AB_10000240 | ICC (1:500) |
| Antibody | Anti-GFP (Rabbit monoclonal) | Abcam | Cat# ab183734, RRID:AB_2732027 | WB (1:30000) |
| Antibody | Anti-GLI1 (Rabbit polyclonal) | Cell Signaling Technology | Cat# 2534, RRID:AB_2294745 | WB (1:500) / ICC (1:100) |
| Antibody | Anti-GLI2 (Goat polyclonal) | R and D systems | Cat# AF3635, RRID:AB_2111902 | WB (1:500) / ICC (1:50) / IHC (1:50) |
| Antibody | Anti-GLI3 (Goat polyclonal) | R and D systems | Cat# AF3690, RRID:AB_2232499 | WB (1:200) / ICC (1:100) / IHC (1:150) |
| Antibody | Anti-gamma-tubulin (Goat polyclonal) | Santa Cruz Biotechnology | Cat# sc-7396, RRID:AB_2211262 | ICC (1:3500) |
| Antibody | Anti-gamma-tubulin (Mouse monoclonal) | Santa Cruz Biotechnology | Cat# sc-17787, RRID:AB_628417 | ICC (1:400) / IHC (1:400) |
| Antibody | Anti-HaloTag (Rabbit polyclonal) | Promega | Cat# G9281, RRID:AB_713650 | ICC (1:700) |

*Continued on next page*

Continued

| Reagent type (species) or resource | Designation | Source or reference | Identifiers | Additional information |
|---|---|---|---|---|
| Antibody | Anti-IFT140 (Rabbit polyclonal) | Proteintech | Cat# 17460–1-AP, RRID:AB_2295648 | ICC (1:100) |
| Antibody | Anti-IFT81 (Rabbit polyclonal) | Proteintech | Cat# 11744–1-AP, RRID:AB_2121966 | ICC (1:50) |
| Antibody | Anti-IFT88 (Rabbit polyclonal) | Proteintech | Cat# 13967–1-AP, RRID:AB_2121979 | ICC (1:100) |
| Antibody | Anti-KATANIN p60 (Mouse monoclonal) | Santa Cruz Biotechnology | Cat# sc-373814, RRID:AB_11014191 | WB (1:1000) |
| Antibody | Anti-KI67 (Rabbit monoclonal) | Abcam | Cat# ab16667, RRID:AB_302459 | ICC (1:500) |
| Antibody | Anti-NPHP1 (Mouse monoclonal) | SIGMA-Aldrich | Cat# MABS2185, N/A | ICC (1:100) |
| Antibody | Anti-mTORC1 (Rabbit monoclonal) | Cell Signaling Technology | Cat# 2972, RRID:AB_330978 | WB (1:1000) |
| Antibody | Anti-NKX2.2 (Mouse monoclonal) | Developmental Studies Hybridoma Bank | Cat# 74.5A5, RRID:AB_531794 | IHC (1:10) |
| Antibody | Anti-NKX6.1 (Mouse monoclonal) | Developmental Studies Hybridoma Bank | Cat# F55A10, RRID:AB_532378 | IHC (1:100) |
| Antibody | Anti-OLIG2 (Rabbit monoclonal) | abcam | Cat# ab109186, RRID:AB_10861310 | IHC (1:500) |
| Antibody | Anti-PAX6 (Mouse monoclonal) | Santa Cruz Biotechnology | Cat# sc-81649, RRID:AB_1127044 | IHC (1:400) |
| Antibody | Anti-Phosho-S6 (Ser 235/236) (Rabbit monoclonal) | Cell Signaling Technology | Cat# 2211, RRID:AB_331679 | WB (1:2000) |
| Antibody | Anti-P-4EBP1 (Thr 37/46) (Rabbit monoclonal) | Cell Signaling Technology | Cat# 2855, RRID:AB_560835 | WB (1:1500) |
| Antibody | Anti-SMO (Mouse monoclonal) | Santa Cruz Biotechnology | Cat# sc-166685, RRID:AB_2239686 | ICC (1:100) |
| Antibody | Anti-SuFu (Mouse monoclonal) | Santa Cruz Biotechnology | Cat# sc-137014, RRID:AB_2197315 | ICC (1:100) |
| Antibody | Anti-S6 (Rabbit monoclonal) | Cell Signaling Technology | Cat# 2217, RRID:AB_331355 | WB (1:2000) |
| Antibody | Anti-4EBP1 (Rabbit monoclonal) | Cell Signaling Technology | Cat# 9644, RRID:AB_2097841 | WB (1:3000) |
| Sequence-based reagent | Human *DYRK2* siRNA#1 | BEX | 608481 | |
| Sequence-based reagent | Human *DYRK2* siRNA#2 | ThermoFisher Scientific | HSS112284 | |
| Sequence-based reagent | Mouse *Dyrk2* siRNA#1 | ThermoFisher Scientific | 4390771 (s87545) | |
| Sequence-based reagent | Mouse *Dyrk2* siRNA#2 | ThermoFisher Scientific | 4390771 (s87546) | |
| Sequence-based reagent | Mouse *Aurka* siRNA#1 | Integrated DNA Technologies | mm.Ri.Aurka.13.1 | |
| Sequence-based reagent | Mouse *Aurka* siRNA#2 | Integrated DNA Technologies | mm.Ri.Aurka.13.4 | |
| Sequence-based reagent | Mouse *Cdc20* siRNA | Integrated DNA Technologies | mm.Ri.Cdc20.13.2 | |
| Sequence-based reagent | Mouse *Kif2c* siRNA | Integrated DNA Technologies | mm.Ri.Kif2c.13.3 | |

*Continued*

| Reagent type (species) or resource | Designation | Source or reference | Identifiers | Additional information |
|---|---|---|---|---|
| Sequence-based reagent | Mouse *Plk1* siRNA | Integrated DNA Technologies | mm.Ri.Plk1.13.1 | |
| Sequence-based reagent | Mouse *Tpx2* siRNA | Integrated DNA Technologies | mm.Ri.Tpx2.13.1 | |
| Sequence-based reagent | Mouse *Ube2c* siRNA | Integrated DNA Technologies | mm.Ri.Ube2c.13.1 | |
| Sequence-based reagent | Negative Control DsiRNA (siNegative) | Integrated DNA Technologies | 51-01-14 | |
| Sequence-based reagent | Silencer Select Negative Control (siControl) | ThermoFisher Scientific | 4390843 | |
| Chemical compound, drug | InSolution SAG | Merck | 566660 | |
| Chemical compound, drug | Rapamycin | LC Laboratories | R-5000 | |
| Software, algorithm | BZ-X800 Analyzer | Keyence | BZ-X800 Analyzer | |
| Software, algorithm | Excel | Microsoft | Mac2019 | |
| Software, algorithm | Fusion | M and S Instruments | Fusion | |
| Software, algorithm | GraphPad Prism 7 | GraphPad Software Inc | Mac OS X | |
| Software, algorithm | PikoReal Software 2.1 | ThermoFisher Scientific | PikoReal Software 2.1 | |

## Generation of *Dyrk2* knockout mice (*Dyrk2*$^{-/-}$)

*Dyrk2*$^{-/-}$ mice were generated using the knockout-first strategy (*Skarnes et al., 2011*). A schematic representation of the targeted *Dyrk2* allele is provided in *Figure 1—figure supplement 1A*. The *Dyrk2* targeting vector (PG00105_X_1_G09 and PG00105_X_1_E04) was obtained from the Knock-out Mouse Project Repository (*Dyrk2* targeting project: 337–66440). Gene-targeting methods were performed according to standard protocols. Briefly, linearized vectors were electroporated into JM8A3.N1 embryonic stem (ES) cells. G418-resistant ES cell clones were analyzed using Southern blot analysis for the presence of the correct targeted-allele using BglII digestion and a 3' external probe. Hybridization with the 3' external probe detected 10.7 kb (wild-type allele) and 17.0 kb (targeted tm1a allele) BglII bands (*Figure 1—figure supplement 1A*). Six positive ES clones out of 240 clones were obtained. Chimeric mice were created by injection of the targeted ES cells into C57BL/6J blastocysts and were mated with C57BL/6J WT mice to establish germline-transmitted founders. Heterozygous knockout-first (*Dyrk2*$^{tm1a}$) mice were identified using Southern blotting. An exon three knockout allele (*Dyrk2*$^{tm1b}$) was generated by mating the *Dyrk2*$^{tm1a}$ mice the with CAG-Cre mice (*Figure 1—figure supplement 1A*). For genotyping and validation of knockout alleles, we performed PCR using the primers listed in *Table 2*.

## Animal care

Mice were housed individually in a temperature-controlled room under a 12 hr light/dark cycle. Determination of pregnancy in mice was achieved by the observation of a vaginal plug on day 0.5 of gestation. Animals were euthanized by anesthesia. The animal experiment protocol was approved by the Institutional Animal Care and Use Committee of Jikei University (No. 2017–065 and 2018–031), and the studies were performed in accordance with the Guidelines for the Proper Conduct of Animal Experiments of the Science Council of Japan.

## Alcian blue and alizarin red staining

Euthanized wild-type and *Dyrk2*$^{-/-}$ mice at E18.5 and E16.5 were skinned, eviscerated, and fixed in 100% EtOH. For skeletal analysis, the embryos were stained with 1% Alcian Blue (Wako Pure

**Table 2.** List of primer sets.

**For genotyping**

| Gene | Sequence (5′→3′) | | Accession number |
|---|---|---|---|
| Dyrk2 tm1b-WT | Forward | TGGGTCCAAATGCAAAGAAACGCCA | NC_000076.6 |
| | Reverse | GCTTCTCGTTCCGCACCATCTTCAG | |
| Dyrk2 tm1b-KO | Forward | CCTTCTCCCTCCTCCACTCTGACCCA | NC_000076.6 |
| | Reverse | CCACACCTCCCCCTGAACCTGAAAC | |

**For amplification of the probes for in situ hybridization or Southern blotting**

| Gene | Sequence (5′→3′) | | Accession number |
|---|---|---|---|
| Mouse Foxf2 | Forward | GAGATTAACCCTCACTAAAGG GAGGTTATGGTGGCCTCGACAT | NM_010225.2 |
| | Reverse | GAGTAATACGACTCACTATAG GGACACACACACCTCCCTTTTCA | |
| Mouse Gli1 | Forward | GAGTATTTAGGTGACACTATAGA AGCAGGGAAGAGAGCAGACTG | NM_010296.2 |
| | Reverse | GAGTAATACGACTCACTATAGGG GCTGAGTGTTGTCCAGGTC | |
| Mouse Ptch1 | Forward | GAGATTAACCCTCACTAAAGGGA CATGGCCTCGGCTGGTAAC | NM_008957.3 |
| | Reverse | GAGTAATACGACTCACTATAGGG TGTACCCATGGCCAACTTCG | |
| Southern for Dyrk2 | Forward | CTTCGAATCCTTTTATCCTTCAGGC | NC_000076.6 |
| | Reverse | ACATCATGTTCATTGGTTTTGCTCT | |

**For cloning**

| Gene | Sequence (5′→3′) | | Accession number |
|---|---|---|---|
| Mouse Aurka CDS | Forward | GGACTCAGATCTCGAGAC ATGGCTGTTGAGGGCG | NM_011497.4 |
| | Reverse | GTCGACTGCAGAATTCC TAAGATGATTTGCTGGTTG | |
| Mouse Dyrk2 CDS | Forward | GTGCGCGATCGCCATGT TAACCAGGAAACCTTCGGC | NM_001014390.2 |
| | Reverse | CTCCGTTTAAACGCTAA CGAGTTTCGGCAACAC | |

**For real-time PCR**

| Gene | Sequence (5′→3′) | | Accession number |
|---|---|---|---|
| Human DYRK2 | Forward | GGGGAGAAAACGTCAGTGAA | NM_006482.3 |
| | Reverse | TCTGCGCCAAATTAGTCCTC | |
| Human HPRT1 | Forward | GGACTAATTATGGACAGGACTG | NM_000194.3 |
| | Reverse | GCTCTTCAGTCTGATAAAATCTAC | |
| Mouse Aurka | Forward | CACACGTACCAGGAGACTTACAGA | NM_011497.4 |
| | Reverse | AGTCTTGAAATGAGGTCCCTGGCT | |
| Mouse Cdc20 | Forward | GAGCTCAAAGGACACACAGC | NM_023223.2 |
| | Reverse | GCCACAACCGTAGAGTCTCA | |
| Mouse Dyrk2 | Forward | CTACCACTACAGCCCACACG | NM_001014390.2 |
| | Reverse | TCTGTCCGTGGCTGTTGA | |
| Mouse Foxf2 | Forward | AGCATGTCTTCCTACTCGTTG | NM_010225.2 |
| | Reverse | TCTTTCCTGTCGCACACT | |
| Mouse Gli1 | Forward | GCACCACATCAACAGTGAGC | NM_010296.2 |
| | Reverse | GCGTCTTGAGGTTTTCAAGG | |
| Mouse Hprt | Forward | CTCATGGACTGATTATGGACAGGAC | NM_013556.2 |
| | Reverse | GCAGGTCAGCAAAGAACTTATAGCC | |

*Table 2 continued on next page*

| | | | |
|---|---|---|---|
| Mouse *Kif2c* | Forward | GAGAGCAAGCTGACCCAGG | NM_134471.4 |
| | Reverse | CCTGGTGAGATCATGGCGATC | |
| Mouse *Plk1* | Forward | CCAAGCACATCAACCCAGTG | NM_011121.4 |
| | Reverse | TGAGGCAGGTAATAGGGAGACG | |
| Mouse *Ptch1* | Forward | CTCTGGAGCAGATTTCCAAGG | NM_008957.3 |
| | Reverse | TGCCGCAGTTCTTTTGAATG | |
| Mouse *Shh* | Forward | GTGAAGCTGCGAGTGACCG | NM_009170.3 |
| | Reverse | CCTGGTCGTCAGCCGCCAGCACGC | |
| Mouse *Tpx2* | Forward | GCGAGGTTGTCAGGTGTGTA | NM_001141977.1 |
| | Reverse | TTGATAAAGTCGGTGGGGGC | |
| Mouse *Ube2c* | Forward | CTGCTAGGAGAACCCAACATC | NM_026785.2 |
| | Reverse | GCTGGAGACCTGCTTTGAATA | |

Chemicals, Osaka, Japan) dissolved in 20% glacial acetic acid and 80% EtOH and 0.01% Alizarin Red (Sigma-Aldrich, St. Louis, MO) dissolved in 1% KOH. The excised tissues were observed using a stereo microscope (BioTools, Gunma, Japan). Ten embryos of each wild-type and *Dyrk2^{-/-}* mice were analyzed.

## In situ hybridization

In situ hybridization was performed according to a previous report (*Fujiwara et al., 2007*). Briefly, each digoxigenin (DIG)-labeled cRNA probe was amplified by PCR using primer sets (*Table 2*) and labeled using the Roche DIG RNA labeling kit (Merck, Schwalbach, Germany). Embryos at E10.5 and the heads of mice at E14.5 were fixed using MEMFA (2 mM EGTA, 1 mM MgSO4, and 3.7% formaldehyde) in 100 mM MOPS (pH 7.5) overnight at 4°C, and this was followed by immersion in 30% trehalose (Wako) in 20 mM HEPES to cryoprotect the tissues. Cryosections (7 μm thickness) from the transverse or sagittal plane were hybridized with DIG-labeled cRNA probe and were visualized with alkaline phosphatase-conjugated anti-DIG antibody (Merck) using 4-nitroblue tetrazolium chloride (NBT; Merck) and 5-bromo-4-chloro-3-indolyl phosphate (BCIP; Merck). The sections were observed under a BZ-X800 microscope (KEYENCE, Osaka, Japan).

## Immunohistochemistry and hematoxylin and eosin (HE)-staining

Embryos at E10.5, E13.5, and E18.5 were fixed and sliced as described above. Depending on the antibody, the sections were antigen retrieved by an ImmunoSaver (Nisshin EM, Tokyo, Japan) for 60 min at 80°C. The sections were incubated with 10% (v/v) fetal bovine serum and 0.4% (v/v) Triton X-100 in HEPES buffer (blocking buffer). After washing, the sections were incubated with primary antibodies (Key resources table) in blocking buffer at 4°C overnight. After the immunoreaction, the sections were incubated with secondary antibodies using Cy3-, Cy5-, or FITC-conjugated AffiniPure donkey anti-goat, rabbit, and mouse IgG (Jackson ImmunoResearch, West Grove, PA). The sections were washed and incubated in VECTASHIELD Mounting Medium (Vector Laboratories, Burlingame, CA) containing 4,6′-diamidino-2-phenylindole dihydrochloride (DAPI). For HE-staining, the sections were stained using standard procedures. The sections were observed under a BZ-X800 fluorescence microscope (KEYENCE).

## Scanning electron microscopy (SEM)

Wild-type and *Dyrk2^{-/-}* embryos at E10.5 were washed with 0.1 M phosphate buffer (PB) (pH7.5) and fixed with 2% glutaraldehyde (TAAB Laboratories Equipment, Berkshire, England) in 0.1 M PB (pH 7.4) for 1 week at 4°C. The embryos were placed in tannic acid in 0.1 M PB for 2 hr at room temperature in darkness, and then immersed in 1% $OsO_4$ solution for 2 hr at room temperature. After dehydration in graded ethanol, the samples were transferred into isoamyl acetate and dried at the critical point in liquid $CO_2$, and this was followed by a metal coating procedure (Hitachi, Tokyo, Japan). The surfaces of tissues were then observed using scanning electron microscopy (Hitachi).

## Plasmid constructs

Full-length cDNA fragments of mouse *Dyrk2* and *Aurka* were amplified by PCR and cloned in frame into the pFN22K-HaloTag-CMVd1-Flexi-vector (Promega, Madison, WI) and pEGFP-C1 (TaKaRa Bio, Otsu, Japan), respectively. The nucleotide sequences of the primers used are listed in *Table 2*.

## Cell culture and transfection

Primary mouse embryonic fibroblast (MEFs) were generated from wild-type and *Dyrk2*$^{-/-}$ embryos at E13.5. MEFs and immortalized human retinal pigment epithelia cells (hTERT-RPE1; Cat# CRL-4000, RRID:CVCL_4388, ATCC, Manassas, VA) were cultured in DMEM (nacalai tesque, Kyoto, Japan) with 10% FBS (biowest, Nuaille, France), 1% GultaMAX (Gibco, Gaithersburg, MD), and 1% Penicillin-streptomycin (nacalai tesque) at 37°C under 5% $CO_2$. hTERT-RPE1 cells were authenticated by the STR profiling and negative for mycoplasma contamination. To induce ciliogenesis, cells were grown to 80–90% confluency and serum-starved (0.5% FBS) for 24 hr. For SAG-stimulation, cells were treated with 100 nM SAG (Merck) for 24 hr after serum-starvation. For rapamycin-stimulation, cells were treated with 0.5 µM rapamycin (LC Laboratories, Woburn, MA) for 24 hr after serum-starvation. Transient knockdown was achieved using the Lipofectamine RNAiMAX transfection regent (Thermo-Fisher Scientific, Waltham, MA) for 48 hr under serum-starvation conditions according to the manufacturer's instructions with a final concentration of 6–20 nM siRNA (Key resources table). For over-expression of DYRK2-HaloTag, transfection was performed using X-tremeGENE9 (Merck) for hTERT-RPE1 cells according to the manufacturer's instructions and the cells were cultured for 24 hr under serum-starvation condition for ciliogenesis. For over-expression of AURKA-EGFP or EGFP, transfection was performed using Xfect (TaKaRa Bio) for MEFs according to the manufacturer's instructions, and the cells were cultured for 24 hr under serum-starvation condition for ciliogenesis.

## Adenovirus infection

Adenovirus construction and infections were performed according to a previous report (*Maekawa et al., 2013*; *Yokoyama-Mashima et al., 2019*). Briefly, Flag-DYRK2 and Flag-DYRK2-K251R (*Taira et al., 2010*; *Mimoto et al., 2013*) were expressed depending upon *Cre*-expression. Following infection at a MOI (multiplicity of infection) of 100, MEFs were extracted for gene-expression analysis at 60 hr post-infection. MOI for MEFs was determined using an adenovirus construct for GFP-expression.

## Immunoblotting

Tissues (the limb buds at E13.5) and MEFs were washed in twice and lysed using RIPA buffer containing several inhibitors (1 mM PMSF, 10 µg/ml Aprotinin, 1 µg/ml Leupeptin, 1 µg/ml Pepstatin A, 1 mM $Na_3VO_4$, 10 mM NaF, and 1 mM DTT). Equal amounts of protein (5 µg) were resolved on 4–15% Mini-PROTEA TGX Precast Protein Gels (BioRad, Hercules, CA). After electrophoresis, proteins were transferred to PVDF membranes (Merck). Membranes were blocked with 5% skim milk in tris-buffered saline (TBS) containing 0.05% Tween 20 (TBST) or 0.1% casein/gelatin in TBST, depending on the antibody. Primary and secondary antibodies were reacted in each blocking buffer (Key resources table). Signals were detected using a chemiluminescent regent, ImmunoStar LD (Wako). Signals were observed and band intensity was measured using a Fusion-Solo system (M and S Instruments, Tokyo, Japan).

## Quantitative real-time polymerase chain reaction (qPCR)

Total RNAs were prepared from tissues (the limbs of E13.5, the mandibular arches of E10.5, and whole embryos at E9.5) and MEFs using the RNeasy mini kit (QIAGEN, Germantown, MD) or ISO-GEN II (Nippon Gene, Tokyo, Japan), respectively. Reverse transcripts were obtained using Prime-Script Reverse Transcriptase (TaKaRa Bio) and subjected to qPCR using a PIKOREAL96 system (ThermoFisher Scientific). Reactions were performed in KAPA SYBR FAST qPCR Master Mix (NIPPON Genetics, Tokyo, Japan) that included 0.2 µM of a specific primer set for each gene (*Table 2*). Data were calculated by the comparative $C_T$ method ($\Delta C_T$ method) to estimate the mRNA copy number relative to that of *Hprt* as an internal standard. The DNA sequence of the PCR product was confirmed by nucleotide sequencing (data not shown).

## Immunocytochemistry

For immunocytochemistry, MEFs and hTERT-RPE1 cells were cultured on 8-well chamber slides (ThermoFisher Scientific) coated with Poly-D-lysin (Sigma-Aldrich). Cells were fixed and antigen retrieved depending on the antibody. The primary antibody reaction was performed at an appropriate dilution (Key resources table) in the presence of blocking buffer at 4°C overnight. After immunoreactions, cells were incubated with secondary antibodies using Cy3-, Cy5-, or FITC-conjugated AffiniPure donkey anti-goat, rabbit, and mouse IgG, and chicken IgY (Jackson ImmunoResearch). The cells were then washed and incubated with DAPI. Immunofluorescence was observed under a BZ-X800 fluorescence microscope (KEYENCE).

## RNA-Seq

Total RNAs were prepared from MEFs cultured in the absence or presence of 100 nM SAG for 24 hr using RNeasy mini kit (QIAGEN) with DNase I treatment (QIAGEN). Materials were enriched for polyA sequences, and quantitative RNA-sequencing was performed using an Illumina HiSeq (Illumina, San Diego, CA). Cutadapt v1.9.1 was used to trim and filter reads, and clean data were aligned to the reference genome (ENSEMBLE, GRCm38.97) using the software HISAT 2 (v2.0.1). Relative gene expression was quantified and normalized in a FPKM (fragments per kilobase of transcript per million mapped reads) format.

## STRING and gene ontology (GO) analysis

To determine the presence of interactions/partnerships among downregulated genes in $Dyrk2^{-/-}$ MEFs, protein-protein interaction networks were extracted from the STRING database (https://string-db.org) and drawn by STRING v11. Gene ontology (GO) analysis was also performed using STRING v11 to demonstrate the biological processes enriched in the altered genes. The resulting GO terms that possessed a false discovery rate (FDR) of less than 0.005 were considered as enriched biological processes.

## Statistical analysis

Each experiment was confirmed by at least three independent biological replicates per condition. Data are presented as the means ± SEM. Prism seven software (GraphPad, San Diego, CA, USA) was used for statistical analyses. Means between two groups were compared using the Student's $t$-test. Multiple inter-group differences were analyzed by one-way ANOVA (analysis of variance) followed by Tukey's multiple comparison test.

# Acknowledgements

We thank Dr. Yuki Kobayashi at Hiroshima University for constructive suggestions. This work was partially supported by JSPS KAKENHI (grant numbers 19K16781 to Saishu Yoshida and 17H03584, 18K19484, and 20H03519 to Kiyotsugu Yoshida), the Platform of Advanced Animal Model Support from MEXT of Japan (to Kiyotsugu Yoshida), the Jikei University Research Fund (to Saishu Yoshida), and the Takeda Science Foundation (to Saishu Yoshida).

# Additional information

### Funding

| Funder | Grant reference number | Author |
| --- | --- | --- |
| Japan Society for the Promotion of Science | 19K16781 | Saishu Yoshida |
| Japan Society for the Promotion of Science | 17H03584 | Kiyotsugu Yoshida |
| Japan Society for the Promotion of Science | 18K19484 | Kiyotsugu Yoshida |
| Japan Society for the Promotion of Science | 16H06276 (AdAMS) | Kiyotsugu Yoshida |

| Takeda Science Foundation | | Saishu Yoshida |
| the Jikei University Research Fund | | Saishu Yoshida |
| Japan Society for the Promotion of Science | 20H03519 | Kiyotsugu Yoshida |

The funders had no role in study design, data collection and interpretation, or the decision to submit the work for publication.

**Author contributions**
Saishu Yoshida, Conceptualization, Data curation, Formal analysis, Funding acquisition, Validation, Investigation, Visualization, Methodology, Writing - original draft, Project administration, Writing - review and editing; Katsuhiko Aoki, Ken Fujiwara, Takashi Nakakura, Akira Kawamura, Satomi Yogosawa, Investigation; Kohji Yamada, Data curation, Visualization; Masaya Ono, Methodology; Kiyotsugu Yoshida, Conceptualization, Supervision, Funding acquisition, Validation, Project administration, Writing - review and editing

**Author ORCIDs**
Saishu Yoshida (iD) https://orcid.org/0000-0002-2574-6410
Akira Kawamura (iD) https://orcid.org/0000-0002-9914-7306
Kiyotsugu Yoshida (iD) https://orcid.org/0000-0003-3108-7383

**Ethics**
Animal experimentation: The animal experiment protocol was approved by the Institutional Animal Care and Use Committee of Jikei University (No. 2017-065 and 2018-031), and the studies were performed in accordance with the Guidelines for the Proper Conduct of Animal Experiments of the Science Council of Japan.

**Decision letter and Author response**
Decision letter https://doi.org/10.7554/eLife.57381.sa1
Author response https://doi.org/10.7554/eLife.57381.sa2

# Additional files

## Supplementary files
• Transparent reporting form

## Data availability
Data except for RNA-seq in this study are included in the manuscript and supporting files. Source data files have been provided: Figure 2-source data 1 Figure 2-figure supplement 1-source data 1 Figure 3-source data 1 Figure 3-source data 2 Figure 3-figure supplement 1-source data 1 Figure 3-figure supplement 2-source data 1 Figure 4-source data 1 Figure 4-figure supplement 1-source data 1 Figure 4-figure supplement 2-source data 1 Figure 4-figure supplement 3-source data 1 Figure 6-source data 1 Figure 6-figure supplement 3-source data 1 Figure 7-source data 1 Figure 8-source data 1 Figure 9-source data 1. RNA-seq data have been deposited in Dryad under accession code URL https://doi.org/10.5061/dryad.pnvx0k6j8.

The following dataset was generated:

| Author(s) | Year | Dataset title | Dataset URL | Database and Identifier |
|---|---|---|---|---|
| Yoshida S, Aoki K, Fujiwara K, Nakakura T, Kawamura A, Yamada K, Ono M, Yogosawa S, | 2020 | The novel ciliogenesis regulator DYRK2 governs Hedgehog signaling during mouse embryogenesis | https://doi.org/10.5061/dryad.pnvx0k6j8 | Dryad Digital Repository, 10.5061/dryad.pnvx0k6j8 |

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
