## [Decision Letter]

Thank you for submitting your article "The novel ciliogenesis regulator DYRK2 governs Hedgehog signaling during mouse embryogenesis" for consideration by *eLife*. Your article has been reviewed by three peer reviewers, one of whom is a member of our Board of Reviewing Editors, and the evaluation has been overseen by Piali Sengupta as the Senior Editor. The reviewers have opted to remain anonymous.

The reviewers have discussed the reviews with one another and the Reviewing Editor has drafted this decision to help you prepare a revised submission.

Summary:

Yoshida et al. characterize the phenotype of *Dyrk2* mutant mice, and show that loss of this kinase results in reduced Hh signaling both in vivo and in cells derived from the *Dyrk2^-/-^* embryos. They also show that *Dyrk2^-/-^* embryos and MEFs have ciliary defects- primarily longer cilia with abnormal tips, and defects in the trafficking of Hh components *Gli2* and *Gli3*. *Dyrk2* is postulated to localize to transition zone of cilia and regulate transcription of AurA, which the authors propose as the mechanism underlying ciliary length changes.

The data presented are robust and thorough. The authors are recognized leaders in understanding the role of *Dyrk2* in cancer pathways and in DNA damage response through phosphorylation of diverse substrates including p53 and c-Jun/Myc. They now highlight a completely novel role of this kinase in embryonic development as a regulator of ciliary morphology and Hh signaling. *Dyrk2* has also been postulated to act as scaffold for EDVP complex in proteolysis of substrates such as the centrosome protein CP110 and microtubule severing enzyme, Katanin. The authors now convincingly show a role of *Dyrk2* as a positive regulator of Hh pathway by careful knockout studies during mouse embryogenesis in the context of craniofacial and skeletal development and Hh signaling assays in vitro. These results are in stark contrast to a previously published paper using *Dyrk2* overexpression that proposed a negative regulatory role of *Dyrk2* in Hh pathway through *Gli2* and Gli3 degradation and direct phosphorylation of *Gli2* at two residues (S385, S1011) (PMID: 18455992). Despite the nicely demonstrated broad role of this kinase in Hh signaling and late embryogenesis, the mechanism proposed for this postulated transition zone-localized protein in causing ciliary dysmorphology through AurA kinase transcription is unclear. Therefore, this work presents interesting findings and is generally well-performed and merits publication, though there are some additional experiments and/or questions to address that could improve the work considerably.

Essential revisions:

1) The authors show low Hh signaling by ISH/qRT PCR of Hh pathway targets in craniofacial region and limbs at e13.5. The later skeletal abnormalities are consistent with Hh signaling defects (although no polydactyly is seen). However, no neural tube patterning defects are seen. Considering that cilia length abnormalities are seen at E10.5 (earlier stages have not been looked at), and *Gli2/3/*SuFu are accumulated in MEF cilia irrespective of *Smo* activation, it is surprising that the Hh signaling defects are not observed in earlier stages of development. The neural tube development defects are shown to be unaffected only at the branchial level at E10.5. The authors should at least test total RNA levels of Hh pathway targets in early stage embryos and rule out dorsoventral patterning defects at earlier stages.

2) The phenotypes of the *Dyrk2^-/-^* mice as described seem to be fairly limited to bone growth and differentiation (and also neural crest-derived craniofacial structures). This certainly points to a role for Hh signaling, but could indicate that the phenotype is tissue specific. Is *Dyrk2* expression tissue restricted? If so this would be very interesting since it would point to *Dyrk2* being a tissue-specific regulator of cilia. Few proteins that regulate cilia in a cell type-specific manner have been identified and little is known about how such regulation is achieved. Making this link might increase the impact of the paper, if it is the case.

3) The localization of *Dyrk2* in transition zone and/or centrosome should be better documented using transition zone markers and necessary controls for the antibody using ko MEFs.

4) The accumulation of *Gli2*/*3* in resting MEF cilia (but not of *Smo*) is similar to PKA null MEFs (PMID: 22007132), which in contrast show high Hh signaling. Here, an alternative hypothesis regarding *Dyrk2* function such as its role in affecting turnaround in ciliary tips and/or affecting axonemal architecture through its function as a kinase for *Gli2* and/or EDVP complex scaffold, respectively might be considered.

5) A more thorough characterization of the ciliary defects in the *Dyrk2^-/-^* cells is desirable. Primarily, what happens to cilia frequency in *Dyrk2^-/-^* cells? If cilia frequency is not affected in these mutants, this would not detract from the work or the conclusions of the paper, but it is important to know the answer. They should also assess whether cilia frequency in cycling *Dyrk2^-/-^* MEFs is different- it seems possible loss of this kinase might increase cilia frequency when few cilia are typically present, especially given that they find that *AurkA* expression is reduced in the mutant cells. Again, it would just be interesting to know this either way.

6) The potential mechanism by which DYRK2 regulates ciliary length is insufficiently discussed/addressed in this study. First of all, the Introduction needs to provide a more thorough and accurate description of the literature relevant for ciliary length control and disassembly, as well as a clear description of what the differences are between steady state ciliary length control and ciliary disassembly observed e.g. during serum re-addition to starved cultures of mammalian cells. Second, the authors suggest that one potential mechanism by which DYRK2 negatively regulates ciliary length is by controlling expression of key cilia disassembly factors such as AURKA, but it is unclear why a transcriptomics approach was used in the first place. Moreover, the observed changes in the transcriptome could be a consequence rather than a cause of the long cilia phenotype seen in DYRK2 deficient cells. Therefore, a rescue experiment that shows normal ciliary length of DYRK2 mutant cells when AURKA expression is normalized must be provided if the authors want to claim that altered AURKA levels are responsible for the phenotype. DYRK2 is a kinase that the authors show is concentrated near the ciliary base; previous work implicated the EDVP complex in regulation of katanin (PMID: 19287380) and CP110 proteolysis (PMID: 28242748). CP110 is a key regulator of ciliogenesis also implicated in ciliary length control, thus an obvious question to ask is what happens to CP110 (centrosomal) levels in cells lacking DYRK2. This experiment should be fairly easy to do as good antibodies against CP110 are commercially available. Katanin levels could be analysed similarly.

7) Subsection “*Dyrk2* deficiency cause suppression of Hedgehog signaling during mouse embryogenesis” and Figure 1: Some quantitative analysis is missing here. How many embryos/animals were examined?

8) Figure 3D: What is the relative expression levels of the wild type and mutant DYRK2 protein in these experiments and are the transfection efficiencies similar for both constructs? This is important to know in order to rule out that the observed difference in rescue effect of the two constructs is not simply due to different cellular expression level. Also, kinase-dead *Dyrk2* does restore significant levels of *Ptch1* transcript with respect to wild-type. Statistical significance for *Gli1* levels is not mentioned with respect to wild-type. *Dyrk2* could have kinase-independent functions as a scaffold.

9) Subsection “DYRK2 regulates ciliogenesis” and Figure 5—figure supplement 3: without quantification the data is not very meaningful, so either the data needs to be quantified or alternatively removed.

10) The manuscript contains several grammatical errors and typos that need to be corrected to enhance readability and clarity.

---

## [Author Response]

Essential revisions:

1) The authors show low Hh signaling by ISH/qRT PCR of Hh pathway targets in craniofacial region and limbs at e13.5. The later skeletal abnormalities are consistent with Hh signaling defects (although no polydactyly is seen). However, no neural tube patterning defects are seen. Considering that cilia length abnormalities are seen at E10.5 (earlier stages have not been looked at), and *Gli2/3/*SuFu are accumulated in MEF cilia irrespective of *Smo* activation, it is surprising that the Hh signaling defects are not observed in earlier stages of development. The neural tube development defects are shown to be unaffected only at the branchial level at E10.5. The authors should at least test total RNA levels of Hh pathway targets in early stage embryos and rule out dorsoventral patterning defects at earlier stages.

As requested, we performed qPCR of Hh target genes using cDNA from whole embryo at E9.5. The results demonstrated a repression of *Gli1* (p=0.077), *Ptch1* (p<0.01), and *Foxf2* (p<0.01) in *Dyrk2^-/-^* embryos newly prepared Figure 2—figure supplement 1B. These data clearly show the defect in Hh signaling in *Dyrk2^-/-^*mice at early stage embryos (E9.5).

To further confirm this finding, we performed in situ hybridization for *Ptch1* at E10.5 newly prepared Figure 2—figure supplement 1C. The results demonstrated that *Ptch1*-expression was decreased in several regions such as the mandibular arch in *Dyrk2^-/-^* embryos, but remained unchanged in the neural tube. Based on these data, we concluded that Hh signaling is active state at the neural tube in *Dyrk2^-/-^* embryos.

Additionally, we speculate that these data have relation to a spatiotemporal expression-pattern of *Dyrk2* in embryo we also commented below (Essential revisions comment 2).

According to the reviewer’s comment, we added newly prepared Figure 2—figure supplement 1B and Figure 2—figure supplement 1C, and their figure legends.

In addition, we added Figure 2—figure supplement 1—source data 1.

We also revised the following sentences in the revised text:

Results: subsection “*Dyrk2* deficiency cause suppression of Hedgehog signaling during mouse embryogenesis”, third paragraph.

Materials and methods: subsection “In situ hybridization” and subsection “Quantitative real-time polymerase chain reaction (qPCR)”.

Table 2: We added information of a primer set for *Ptch1*-probe.

2) The phenotypes of the *Dyrk2^-/-^* mice as described seem to be fairly limited to bone growth and differentiation (and also neural crest-derived craniofacial structures). This certainly points to a role for Hh signaling, but could indicate that the phenotype is tissue specific. Is *Dyrk2* expression tissue restricted? If so this would be very interesting since it would point to *Dyrk2* being a tissue-specific regulator of cilia. Few proteins that regulate cilia in a cell type-specific manner have been identified and little is known about how such regulation is achieved. Making this link might increase the impact of the paper, if it is the case.

We appreciate your important suggestion and really agree with your comment.

We have tried to identify the localization of DYRK2 using wild-type and *Dyrk2^-/-^* embryos by immunohistochemistry and in situ hybridization. Indeed, we have performed the immunohistochemistry using almost all commercially available antibodies against DYRK2 (more than 15 antibodies). In addition, we have tried to develop original antibodies using two independent epitopes.

Although we have also verified various conditions (e.g. fixation- and antigen retrieved-conditions using cryo-, frozen-, and paraffin-sections), we could not successfully detect specific immuno-positive signals (i.e. detecting only wild-type but not *Dyrk2^-/-^* embryos). We have also performed in situ hybridization using several probes for *Dyrk2*; however, we could not obtain positive results.

We would thus argue that, in the future, we would like to try to show it in a separate study.

3) The localization of *Dyrk2* in transition zone and/or centrosome should be better documented using transition zone markers and necessary controls for the antibody using ko MEFs.

As suggested, we performed immunocytostaining for a TZ marker, NPHP1, in hTERT-RPE1 cells transfected with DYRK2-Halo vector. The results demonstrated that DYRK2-Halo-tag-positive signals are co-localized in NPHP1-positive signals newly prepared Figure 5D.

We analyzed the cellular localization of DYRK2 using hTERT-RPE1 cells overexpressed DYRK2-Halo vector, because we could not find available anti-DYRK2 antibodies to detect endogenous DYRK2 by immunocytostaining. Therefore, we performed a control experiment for anti-Halo tag and DYRK2 antibody in hTERT-RPE1 cells transfected with “empty vector (pFN22K-Halo Tag-CMVd1-Flexi-vector)”. The results indicated that no signal was observed newly prepared Figure 5C.

Accordingly, we added newly prepared Figure 5C and D and their figure legends.

We also added the following sentences in the revised text: “No signal for anti-HaloTag (Figure 5C) or anti-DYRK2 (data not shown) was observed in hTERT-RPE1 cells transfected with empty vector (pFN22K-Halo Tag-CMVd1-Flexi-vector). Moreover, immuno-positive signals for DYRK2-HaloTag were co-localized with a TZ marker, NPHP1 (Figure 5D)”.

We added information of anti-NPHP1 antibody and pFN22K-Halo Tag-CMVd1-Flexi-vector (empty vector) in Key Resources Table.

4) The accumulation of *Gli2/3* in resting MEF cilia (but not of *Smo*) is similar to PKA null MEFs (PMID: 22007132), which in contrast show high Hh signaling. Here, an alternative hypothesis regarding *Dyrk2* function such as its role in affecting turnaround in ciliary tips and/or affecting axonemal architecture through its function as a kinase for *Gli2* and/or EDVP complex scaffold, respectively might be considered.

Thank you for your important suggestion. As our response to Essential revisions comment 6, to identify DYRK2’s substrate involving in ciliogenesis, we have analyzed protein levels of CP110 and KATANIN p60, which are postulated as DYRK2’s substrates, in *Dyrk2^-/-^*MEFs. The results showed no obvious difference between wild-type and *Dyrk2^-/-^*MEFs in these proteins newly prepared Figure 6—figure supplement 4.

While we are now trying to identify novel substrates of DYRK2 for ciliogenesis, we need more time to accomplish. In this context, we would like to show it in a separate study regarding the identification of novel substrates of DYRK2 involving in ciliogenesis.

Accordingly, we added newly prepared Figure 6—figure supplement 4 and its figure legend.

We also added the following sentence in the revised text: “Moreover, a centrosome protein CP110 ,(Maddika and Chen 2009) and a microtubule severing enzyme, KATANIN p60 et al.,(Hossain 2017), have been identified as substrates of DYRK2 for proteolysis. In *Dyrk2^-/-^* MEFs, however, no obvious difference in protein levels of both CP110 and KATANIN p60 was observed (Figure 6—figure supplement 4)”.

We added information of anti-CP110 and KATANIN p60 antibodies in Key Resources Table.

5) A more thorough characterization of the ciliary defects in the *Dyrk2^-/-^* cells is desirable. Primarily, what happens to cilia frequency in *Dyrk2*^*-/*-^ cells? If cilia frequency is not affected in these mutants, this would not detract from the work or the conclusions of the paper, but it is important to know the answer. They should also assess whether cilia frequency in cycling *Dyrk2^-/-^* MEFs is different- it seems possible loss of this kinase might increase cilia frequency when few cilia are typically present, especially given that they find that *AurkA* expression is reduced in the mutant cells. Again, it would just be interesting to know this either way.

According to the reviewer’s comment, we measured the proportion of ciliated cells newly prepared Figure 4—figure supplement 3A-B, and confirmed no difference between wild-type and *Dyrk2^-/-^*MEFs.

We also analyzed the proportion of ciliated cells in cell cycling wild-type and *Dyrk2^-/-^*MEFs using KI67-staining newly prepared Figure 4—figure supplement 3C. The results showed that both wild-type and *Dyrk2^-/-^*MEFs in KI67-positive cells were hardly ciliated.

We also added the following sentences in the revised text: “In contrast to the length and morphology of primary cilia, no difference was observed on the proportion of ciliated cells in wild-type and *Dyrk2^-/-^*MEFs (Figure 4—figure supplement 3A-B). Similarly, in cell-cycling (KI67-positive) wild-type and *Dyrk2^-/-^*MEFs, there was comparable in the proportion of ciliated cells (ciliated cells in KI67-positive cells is 1 per 199 and 1 per 139 cells in wild-type and *Dyrk2^-/-^*MEFs, respectively) (Figure 4—figure supplement 3C)”.

We added Figure 4—figure supplement 3—source data 1, and information of anti-KI67 and ARL13B (abcam) antibodies in Key Resources Table.

6) The potential mechanism by which DYRK2 regulates ciliary length is insufficiently discussed/addressed in this study. First of all, the Introduction needs to provide a more thorough and accurate description of the literature relevant for ciliary length control and disassembly, as well as a clear description of what the differences are between steady state ciliary length control and ciliary disassembly observed e.g. during serum re-addition to starved cultures of mammalian cells.

As indicated, we revised the third paragraph of the Introduction.

Second, the authors suggest that one potential mechanism by which DYRK2 negatively regulates ciliary length is by controlling expression of key cilia disassembly factors such as AURKA, but it is unclear why a transcriptomics approach was used in the first place. Moreover, the observed changes in the transcriptome could be a consequence rather than a cause of the long cilia phenotype seen in DYRK2 deficient cells. Therefore, a rescue experiment that shows normal ciliary length of DYRK2 mutant cells when AURKA expression is normalized must be provided if the authors want to claim that altered AURKA levels are responsible for the phenotype. DYRK2 is a kinase that the authors show is concentrated near the ciliary base; previous work implicated the EDVP complex in regulation of katanin (PMID: 19287380) and CP110 proteolysis (PMID: 28242748). CP110 is a key regulator of ciliogenesis also implicated in ciliary length control, thus an obvious question to ask is what happens to CP110 (centrosomal) levels in cells lacking DYRK2. This experiment should be fairly easy to do as good antibodies against CP110 are commercially available. Katanin levels could be analysed similarly.

First of all, we have also focused on and analyzed protein levels of CP110 and KATANIN p60, which are postulated as DYRK2’s substrates. Immuno-blotting of CP110 and KATANIN p60, however, showed no obvious difference between wild-type and *Dyrk2^-/-^*MEFs newly prepared Figure 6—figure supplement 4.

Therefore, to understand the phenotype of *Dyrk2^-/-^* cells in ciliogenesis in more detail and find novel clues, we performed transcriptome analysis.

Second, we performed a rescue experiment by over-expression of AURKA-EGFP in *Dyrk2^-/-^*MEFs newly prepared Figure 9. We observed a reduction of the cilia length in AURKA-EGFP-transfected *Dyrk2^-/-^* MEFs, but not in EGFP-transfected *Dyrk2^-/-^* MEFs. These data support that downregulated *Aurka-*expression is, at least in part, associated with phenotypes of *Dyrk2^-/^*^-^ cells.

Accordingly, we revised following points,

Related to CP110 and KATANIN p60:

We added newly prepared Figure 6—figure supplement 4 and its figure legend.

We also revised the text subsection “Deletion of DYRK2 induces abnormal ciliary trafficking of Hedgehog pathway components”, last paragraph.

We added information of anti-CP110 and KATANIN p60 antibody in Key Resources Table.

Related to a rescue experiment of AURKA:

We added new data newly prepared Figure 9 and its figure legend.

In addition, we added Figure 9—source data 1.

We also revised the following sentences in the revised text:

Results: subsection “Deletion of *Dyrk2* dysregulates the expression of *Aurka* and other cilia-disassembly genes”, end of first paragraph.

Materials and methods: subsection “Plasmid constructs”, subsection “Cell culture and transfection”, and subsection “Immunocytochemistry”.

Table 2: We added information of a primer set for mouse *Aurka* CDS.

We added information of expression vector (mouse *Aurka*/pEGFP-C1 and pEGFP-C1) anti-GFP antibodies in Key Resources Table.

7) Subsection “*Dyrk2* deficiency cause suppression of Hedgehog signaling during mouse embryogenesis” and Figure 1: Some quantitative analysis is missing here. How many embryos/animals were examined?

According to the reviewer’s comment, we added the sentence: “Ten embryos of each wild-type and *Dyrk2^-/-^* mice were analyzed”.

8) Figure 3D: What is the relative expression levels of the wild type and mutant DYRK2 protein in these experiments and are the transfection efficiencies similar for both constructs? This is important to know in order to rule out that the observed difference in rescue effect of the two constructs is not simply due to different cellular expression level.

We added the immune-blotting data showing over-expression protein levels newly prepared Figure 3—figure supplement 1D and its figure legend, and confirmed that both WT and *K251R* were expressed with equal levels.

Additionally, to describe a method for determination of MOI, we added the following sentence “MOI for MEFs was determined using an adenovirus construct for GFP-expression” in the revised text, and information of an adenovirus for GFP-expression in Key Resources Table.

Also, kinase-dead *Dyrk2* does restore significant levels of *Ptch1* transcript with respect to wild-type. Statistical significance for *Gli1* levels is not mentioned with respect to wild-type. *Dyrk2* could have kinase-independent functions as a scaffold.

First, we added the *P*-value “p=0.254” in Figure 3D. As pointed out, kinase-dead DYRK2 (*K251R*) showed a moderate effect for expression of *Gli1* and *Ptch1*. We consider that these data show a kinase independent function of DYRK2 (e.g. as scaffold protein, as the reviewers pointed out). Therefore, we added the following sentences “Additionally, over-expression of the *K251R* construct slightly increased *Gli1* and *Ptch1* expression in comparison with that of empty vector (Figure 3D). This kinase-independent effect might be associated with a function of DYRK2 as a scaffold protein Maddika and Chen, 2009”.

9) Subsection “DYRK2 regulates ciliogenesis” and Figure 5—figure supplement 3: without quantification the data is not very meaningful, so either the data needs to be quantified or alternatively removed.

According to the reviewer’s comment, we removed the data Original Figure 5—figure supplement 3: Stability of primary cilia in *Dyrk2^-/-^*MEFs. Accordingly, we deleted the following sentences from the revised text: “Although we confirmed the acetylated and glutamylated tubulin modifications in the axoneme that are associated with stability of microtubules ,(Janke and Bulinski 2011), both modifications remained unchanged in *Dyrk2^-/-^*MEFs cilia according to immunostaining. Additionally, destabilization of axonemal microtubules induced by exposure to 4 °C et al.,(He 2014) showed no difference in loss of acetylated tubulin modification compared to that observed in wild-type”.

We also deleted an information of anti-Glutamylated tubulin antibody from Key Resources Table.

10) The manuscript contains several grammatical errors and typos that need to be corrected to enhance readability and clarity.

As requested, we carefully checked grammatical errors and typos throughout the revised manuscript and appropriately revised.